# EGOC inhibits TOROID polymerization by structurally activating TORC1

Manoël Prouteau ®[1,7] ✉, Clélia Bourgoint ®[1,7], Jan Felix ®[2,3,7] ✉, Lenny Bonadei[1,7], Yashar Sadian[4], Caroline Gabus[1], Savvas N. Savvides ®[2,3], Irina Gutsche ®[5], Ambroise Desfosses ®[5] & Robbie Loewith ®[1,6] ✉

Target of rapamycin complex 1 (TORC1) is a protein kinase controlling cell homeostasis and growth in response to nutrients and stresses. In *Saccharomyces cerevisiae*, glucose depletion triggers a redistribution of TORC1 from a dispersed localization over the vacuole surface into a large, inactive condensate called TOROID (TORC1 organized in inhibited domains). However, the mechanisms governing this transition have been unclear. Here, we show that acute depletion and repletion of EGO complex (EGOC) activity is sufficient to control TOROID distribution, independently of other nutrient-signaling pathways. The 3.9-Å-resolution structure of TORC1 from TOROID cryo-EM data together with interrogation of key interactions in vivo provide structural insights into TORC1-TORC1' and TORC1-EGOC interaction interfaces. These data support a model in which glucose-dependent activation of EGOC triggers binding to TORC1 at an interface required for TOROID assembly, preventing TORC1 polymerization and promoting release of active TORC1.

The target of rapamycin (TOR) is a Ser/Thr protein kinase and a member of the phosphatidylinositol kinase-related protein kinase family[1]. TOR functions as the catalytic subunit of distinct multiprotein complexes, rapamycin-sensitive TORC1 and rapamycin-insensitive TORC2. Both are broadly conserved across eukaryotes, functioning as essential regulators of cell homeostasis[2,3]. In yeast, TORC1 controls cell mass and volume according to nutrient availabilities and abiotic stresses, whereas TORC2 maintains biophysical properties of the plasma membrane[4,5].

Yeast TORC1 is composed of either of the two TOR paralogs Tor1 or Tor2, in addition to Kog1, Lst8 and the yeast-specific subunit Tco89 (refs. [6,7]). In humans, mTORC1 (mammalian TORC1) contains mTOR, Raptor and mLst8. TORC1 phosphorylates the AGC-family kinase Sch9, which serves as a reporter of TORC1 activity in the presence of good nutrient sources and inhibition upon nutrient deprivation as well as exposure to osmotic, redox or thermal stresses[8].

TORC1 effectors regulate a plethora of molecular processes ranging from ribosome biogenesis to autophagy and from lipid biosynthesis to membrane trafficking[9–12]. mTORC1 is also regulated downstream of growth-factor signaling and phosphorylates the AGC-family kinase S6K1. Compromised mTORC1 regulation has been implicated in several illnesses including cancers, neurodegenerative diseases and metabolic disorders[13].

X-ray crystallography and cryo-EM[14–16] studies have demonstrated that mTORC1 is dimeric and elucidated how rapamycin recruits the proline isomerase FKBP12 to sterically occlude the kinase active site. Multiple GTPases regulate mTORC1 activity. The Rheb GTPase couples growth-factor and energy cues to mTORC1, with binding of GTP-loaded Rheb leading to allosteric rearrangements that increase kinase activity of mTORC1 (ref. [16]). The role of RAG GTPases, downstream of nutrient cues, is more nuanced. There are two sets of *RAG* paralogues, *RAGA/B*, and *RAGC/D*. They form obligate heterodimers,

[1]Department of Molecular and Cellular Biology, University of Geneva, Geneva, Switzerland. [2]Unit for Structural Biology, Department of Biochemistry and Microbiology, Ghent University, Ghent, Belgium. [3]Unit for Structural Biology, VIB-UGent Center for Inflammation Research, Ghent, Belgium. [4]CryoGEnic facility (DCI Geneva), University of Geneva, Geneva, Switzerland. [5]Institut de Biologie Structurale, Université Grenoble Alpes, CEA, CNRS, IBS, Grenoble, France. [6]Swiss National Centre for Competence in Research Chemical Biology, Geneva, Switzerland. [7]These authors contributed equally: Manoël Prouteau, Clélia Bourgoint, Jan Felix, Lenny Bonadei. ✉e-mail: manoel.prouteau@unige.ch; Jan.Felix@UGent.be; robbie.loewith@unige.ch

with GTP-loading status determining the distance and relative orientation between the two G-domains[17,18]. The RAGs are anchored to the lysosome membrane via the heteropentameric Ragulator complex[19–21]. Recent structures of the Raptor–RAG–Ragulator complex suggest that active RagA$^{GTP}$–RagC$^{GDP}$ is able to bind the Armadillo domain of Raptor and grasp a specific short α-helical fragment termed the 'claw'[22,23]. This binding recruits mTORC1 to the lysosome, where it can be activated by Rheb$^{GTP}$.

In contrast to mammalian cells and *Schizosaccharomyces pombe*[24], the yeast Rheb orthologue, Rhb1, does not appear to contribute to TORC1 regulation. Furthermore, yeast TORC1 appears to be constitutively localized to vacuole and/or endosome membranes, regardless of its activity status[25,26]. Acute glucose starvation triggers the condensation of TORC1 into a large, vacuole-associated helical polymer named TOROID[27]. TOROID formation leads to Sch9 dephosphorylation, implying that this polymerization is responsible for TORC1 inactivation. Consistently, a low-resolution TOROID structure has suggested that stacking of TORC1 sterically occludes the kinase active site, analogous to rapamycin-FKBP12 binding[27]. TOROID formation was found to be inhibited by the two yeast RAG orthologues Gtr1 (RAGA/B) and Gtr2 (RAGC/D), because Δ*gtr1* Δ*gtr2* cells form constitutive TOROIDs, even in the presence of glucose, and do not downregulate Sch9 phosphorylation upon glucose starvation[27].

Gtr1 and Gtr2 bind to the EGO ternary complex (EGO-TC) composed of Ego1, Ego2 and Ego3 to form EGOC, which is tethered to the vacuolar membrane via myristoylation and palmitoylation of the Ego1 amino-terminal extremity[28]. Ego1, Ego2 and Ego3 cradle the carboxyl-terminal dimerization domains of Gtr1 and Gtr2 (ref. [29]) in a manner that presents the N-terminal G-domains to regulators and effectors. The GTPase-activating proteins (GAPs) that act on Gtr1 and Gtr2 are SEACIT and Lst4–7, respectively[30,31]. Although numerous guanine nucleotide exchange factors that act on Gtr1 have been reported[18,25,31], Gtr2 may not require a guanine nucleotide exchange factor, as it possesses an intrinsically high dissociation constant for GDP[32]. The guanine nucleotide-loading status of the Gtrs appears to have a direct effect on TORC1 localization and activity regulation in yeast. In cells expressing 'inactive' Gtr1$^{GDP}$–Gtr2$^{GTP}$, a TOROID punctum on the vacuolar membrane is prominent, and TORC1 activity is reduced; in cells expressing 'active' Gtr1$^{GTP}$–Gtr2$^{GDP}$, TORC1 is diffusely localized over the vacuolar surface, and its activity is high[26,27]. However, the molecular details of how the EGOC governs TOROID assembly, alone or in conjunction with other nutrient-regulated signaling pathways, remain unknown.

To assess the contribution of the EGOC to TORC1 localization dynamics, independently of other signaling pathways[33], we developed orthogonal approaches to acutely abrogate or reintroduce EGOC activity. Proteolytic shaving of the EGOC from the vacuole membrane triggered TORC1 puncta (TOROID) formation, whereas reintroduction of the EGOC to the vacuole membrane triggered puncta disassembly. We also observed that the EGOC colocalized with TORC1 puncta in a Gtr-dependent fashion. To better characterize this interaction, we used cryo-EM to obtain a 3.9-Å-resolution structure of TORC1 from TOROID cryo-EM maps. We found that a central hub of interactions locks the Kog1 Armadillo domain on top of the kinase domain of Tor2 (Tor2′) in the neighboring protomer/dimer in the TOROID helix (such interprotomer interactions are indicated with a prime throughout) to sterically occlude the kinase active site. CRISPR–Cas9 mutagenesis and pull-down assays were used to verify these core features and revealed that this same hub is engaged by EGOC to activate TORC1 signaling in response to glucose. Based on these data, we propose that the EGOC is necessary and sufficient to prevent TOROID assembly, by competing for a common binding hub; binding of the Kog1 hub to Tor1′/2′ sequesters TORC1 into an inactive TOROID, whereas binding to active EGOC liberates TORC1 dimers that are able to signal to downstream targets.

## Results

### EGOC activity change affects TOROID independently of glucose

Based on phenotypes of Δ*gtr1* Δ*gtr2* cells, and cells expressing GTP/GDP-locked Gtr1/2, we previously proposed that TOROID regulation downstream of glucose cues was mediated via the EGOC[27]. However, glucose signals also regulate TORC1 activity via additional, parallel pathways including Snf1/AMPK and pH[34,35]. Therefore, we wished to determine whether orthogonal, glucose-independent manipulation of EGOC function alone would alter TOROID dynamics.

To rapidly inactivate EGOC, we introduced a tobacco etch virus (TEV) protease cleavage site into the N-terminal region of Ego1, downstream of the lipidation sites (Ego1$^{TEV}$; Fig. 1a and Extended Data Fig. 1a)[28,36]. TEV protease was expressed downstream of the *CTH2* promoter, whose tight repression by Fe$^{2+}$ is quickly reversed by addition of the iron chelator bathophenanthrolinedisulfonic acid (BPS)[37]. Neither acute BPS treatment nor TEV protease induction affected TORC1 activity in otherwise wild-type (WT) cells (Extended Data Fig. 1b).

Although the *EGO1$^{TEV}$* allele does not display a phenotype, expression of the TEV protease renders these cells hypersensitive to rapamycin (Extended Data Fig. 1c)[25,38]. In *EGO1$^{TEV}$* cells, TEV expression diminishes the EGOC vacuolar localization by 50% after 30 min, and almost completely after 75 min (Fig. 1b and Extended Data Fig. 1d). Shaving EGOC from the vacuolar membrane doubled the number of cells presenting TOROIDs after ~60 min (Fig. 1c,d). Single-cell measurements estimated the half-time of TOROID formation following EGOC depletion to be about 7 min (Fig. 1e,f).

To rapidly reactivate EGOC, we exploited the vacuole fusion that occurs during zygote formation[39]. We crossed a *MATa* Δ*gtr1* Δ*gtr2* strain, which constitutively displays a TOROID marked with GFP-Tor1, with a *MATα* partner containing Vph1-mCherry to delineate the vacuolar membrane (Fig. 1g); a *MATα* Δ*gtr1* Δ*gtr2 VPH1-mCherry* strain served as a control. Budding zygotes were staged as either early, intermediate or late (Fig. 1g). In crosses between Δ*gtr1* Δ*gtr2* parents, a TOROID was detected in about 50% of buds and in some *MATα* parents already at the intermediate zygote stage (Fig. 1g,h). This was not observed in crosses with the *GTR1 GTR2* partner. In this case, TORC1 puncta were observed in only very few buds and at late stages even disappeared from the *MATa* parent (Fig. 1h). In live, single-cell imaging, TOROIDs from this mating were observed to dissolve promptly upon entry into the bud, that is, upon encounter with functional EGOC (half-time ~2.5 min; Fig. 1i,j).

These data indicate that acute loss or reestablishment of EGOC activity, in the absence of other confounding nutrient-regulated pathways, is sufficient to respectively assemble or dissolve TOROIDs. This supports the notion that the EGOC regulates TOROID formation, potentially through a previously described physical interaction with TORC1 (refs. [25,40]) and/or TOROIDs.

### TOROIDs partially colocalize with EGOC puncta

We assessed how EGOC and TORC1/TOROIDs localize with respect to one another. TORC1 (*GFP-TOR1*) and EGOC (*EGO3-3xmCherry*) have been reported to colocalize in discrete puncta[41,42]. However, in this background these alleles displayed a synthetic interaction, as TORC1 puncta are present even in glucose-replete conditions (Extended Data Fig. 2a). Importantly, when assessed separately, both TORC1 (*GFP-KOG1*) and the EGOC (*EGO3-GFP*) localized diffusely around the vacuole membrane in glucose-replete cells but collapsed into a punctum within 10 min of acute glucose depletion. Both puncta dissolved again within minutes of glucose replenishment (Fig. 2a,b).

The *GFP-KOG1 EGO3-3xmCherry* strain was relatively uncompromised, with levels of TOROIDs in glucose-replete and postdiauxic shift (PDS) cultures close to those seen in singly tagged strains (Fig. 2c and Extended Data Fig. 2a). In PDS cultures, colocalization between TORC1 and EGOC was moderate (Pearson's coefficient (PC): 0.44; Extended Data Fig. 2b) and asymmetric; whereas EGOC overlapped strongly with

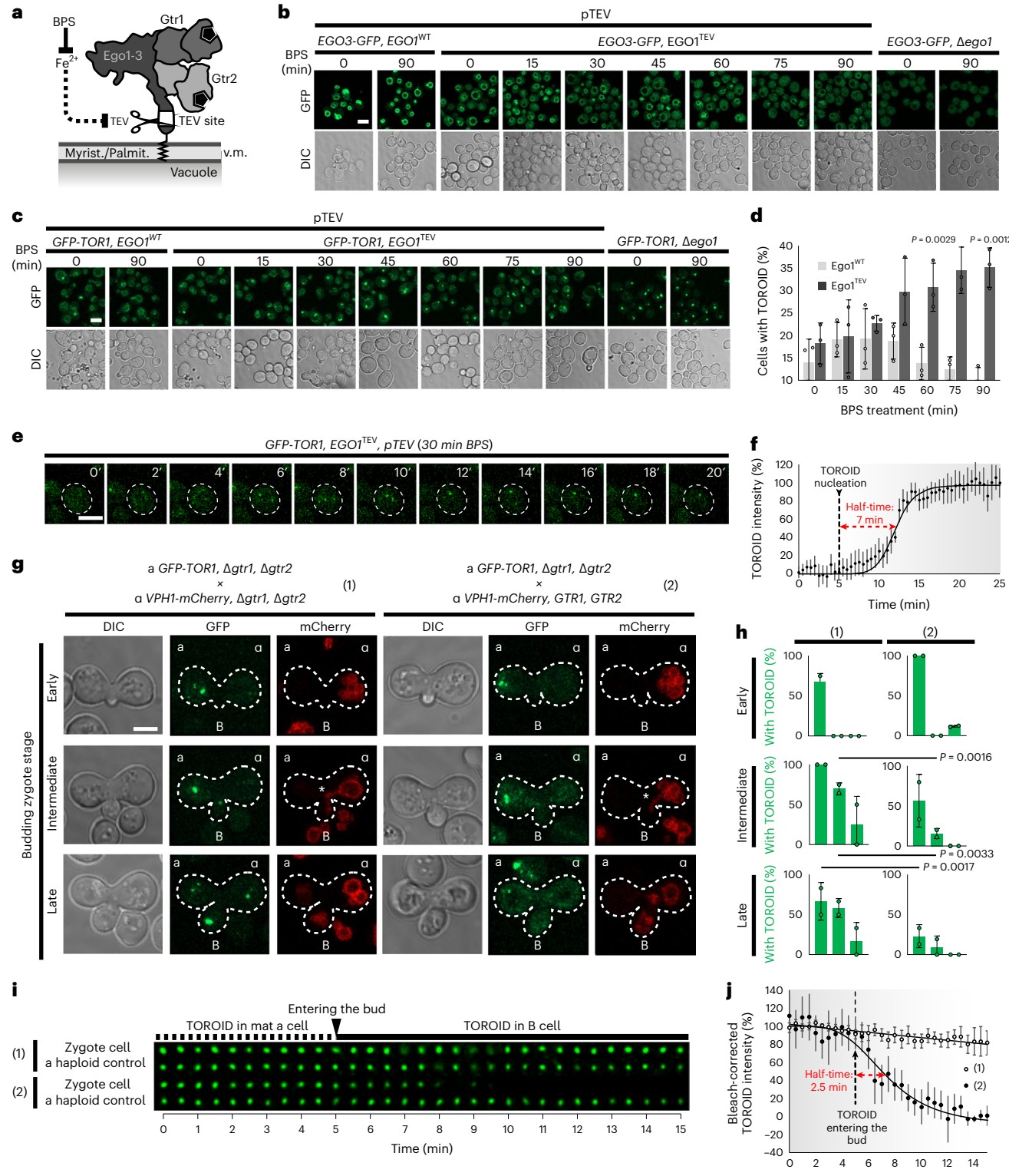

**Fig. 1 | Acute removal of the EGOC from the vacuole membrane and its restoration induce TOROID formation and dissolution, respectively.**
**a**, Schematic depicting the strategy used to shave EGOC from the vacuole membrane. **b**, EGOC (*EGO3-GFP*) localization in Ego1WT and Ego1TEV strains following TEV protease induction and in an Δ*ego1* strain as control lacking vacuolar localization. **c**, TORC1 (*GFP-TOR1*) monitored in the strains described in **b**. **d**, Quantification of data represented in c from *n* = 3 independent experiments with at least 171 cells examined per condition. **e**, Single-cell analyses after 30 min of TEV induction in *GFP-TOR1, ego1TEV*. Time lapses in minutes. **f**, Relative intensity of the TOROIDs extracted from the data represented in **e** from *n* = 4 independent experiments with at least three cells analyzed per replicate. **g**, TOROID dynamics

following EGOC restoration in three mating stages: early, intermediate and late. *MATa GFP-TOR1* strain (a) is crossed with *MATalpha* mating *VPH1-mCherry* strain (α), which carries either (1) Δ*gtr1 gtr2* or (2) *GTR1 GTR2*. B indicates the bud of the zygote. **h**, Percentages of a, α and B cells containing a TOROID from data presented in **g** from *n* = 2 independent experiments with at least 13 zygotes tracked per replicate. **i**, Kymographs tracking TOROID dynamics upon matings in **g**. α cells not engaged in mating are used as bleaching controls. **j**, Quantification of data represented in **i** from *n* = 4 independent experiments with at least two zygotes analyzed per replicate. Scale bars, 5 µm. Myrist., myristoylation; Palmit., palmitoylation; v.m., vacuole membrane; DIC, Differential interference contrast microscopy.

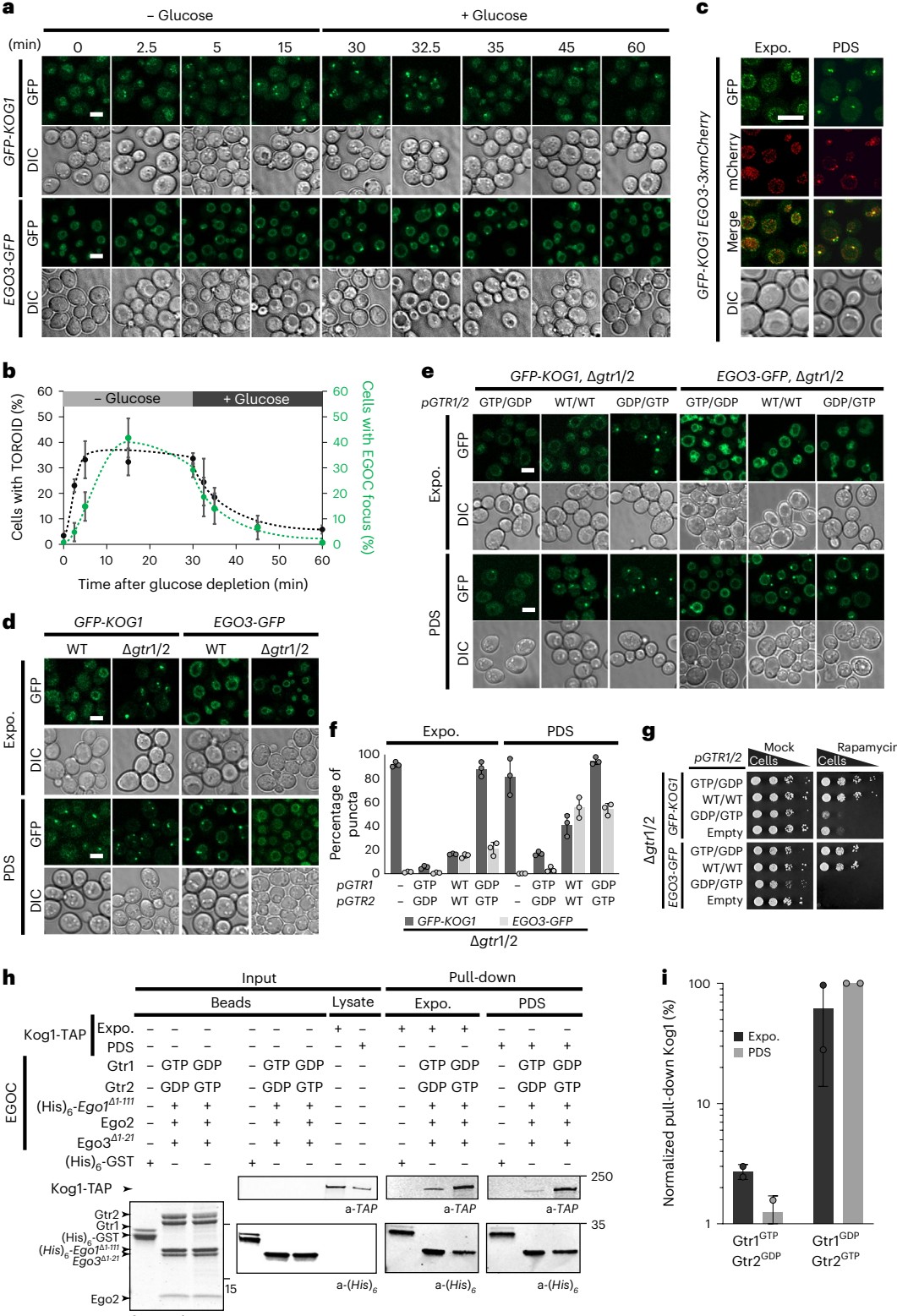

**Fig. 2 | EGOC and TORC1 form colocalized and interdependent puncta.**
**a**, *GFP-KOG1* and *EGO3-GFP* cells following glucose depletion and subsequent readdition. **b**, Plot representing the quantification of puncta formation from **a** from at least *n* = 2 independent experiments with a minimum of 66 cells examined per replicate. In **c**–**e**, cells were imaged in exponential phase or PDS. **c**, *GFP-KOG1 EGO3-3xmCherry* cells. **d**, *GFP-KOG1* or *EGO3-GFP*, in WT or *Δgtr1 Δgtr2* cells. **e**, *GFP-KOG1* or *EGO3-GFP Δgtr1 Δgtr2* cells were cotransformed with *pGTR1^{WT} pGTR2^{WT}*, *pGTR1^{GTP} pGTR2^{GDP}* or *pGTR1^{GDP} pGTR2^{GTP}* plasmids. **f**, Bar graph representing the quantification of puncta formation (TOROID and EGOC) from **e**

from *n* = 2 independent experiments with at least 77 cells examined per replicate. **g**, Cells with the indicated genotypes spotted onto plates containing 2.5 nM rapamycin. **h**, In vitro pull-down of TORC1, purified from exponential or PDS cultures, using recombinant EGOC containing either 'active' (Gtr1^{GTP}–Gtr2^{GDP}) or 'inactive' (Gtr1^{GDP}–Gtr2^{GTP}) GTPase dimer. (His)$_6$-purified proteins are shown in Coomassie-stained SDS–PAGE. Input and pull-down are shown by western blotting using primary antibodies to detect the corresponding proteins. **i**, Quantification of the relative TORC1 pull-down efficiency from **h** with *n* = 3. Scale bars, 5 μm. Expo., exponential phase; PDS, post diauxic shift.

TORC1 (Mander's coefficient (MC): 0.71; Extended Data Fig. 2b), TORC1 overlap with EGOC was less robust (MC: 0.45; Figure Extended Data Fig. 2b). Small EGOC puncta did not colocalize with TORC1 ($PC_{<500 nm}$: −0.004 ± 0.095; Extended Data Fig. 2c), whereas large EGOC puncta did ($PC_{>500 nm}$: 0.525 ± 0.175). By contrast, large and small TORC1 puncta displayed similar overlaps with EGOC ($PC_{<500 nm}$: 0.315 ± 0.023; $PC_{>500 nm}$: 0.361 ± 0.095; Extended Data Fig. 2c). Together, these data show that both EGOC and TORC1 form condensates upon acute glucose depletion or in PDS cells, and that large EGOC puncta seem to require a TOROID but not vice versa.

We characterized how guanine nucleotide-loading status of the Gtrs affected EGOC puncta. Although deletion of *GTRs* promotes TORC1 puncta formation (Prouteau et al.[27], Fig. 1), EGO-TC puncta were absent in this strain (Fig. 2d). Reintroduction of 'active' *GTR* alleles (Gtr1[GTP]–Gtr2[GDP]) antagonized both TORC1 and EGOC puncta formation regardless of the growth state, whereas reintroduction of 'inactive' *GTR* alleles (Gtr1[GDP]–Gtr2[GTP]) enhanced formation of TOROIDs but, curiously, not EGOC puncta (Fig. 2e,f). In Δ*gtr1* Δ*gtr2* and Gtr1[GDP]–Gtr2[GTP] strains, which contain TOROIDs, TORC1 activity was reduced as evidenced by their rapamycin hypersensitivity (Fig. 2g). As a complementary approach, we used mutant strains in which Gtr GAP activities had been altered. *IML1/SEA1 deletion*, which leads to Gtr1[GTP] accumulation, antagonized EGOC and TORC1 puncta (Extended Data Fig. 2d–f), whereas deletion of *SEA3* or *LST4–LST7*, which lead to Gtr1[GDP] or Gtr2[GTP] accumulation, respectively, augmented EGOC and TORC1 puncta in exponentially growing cells (Extended Data Fig. 2d–f). Collectively, these results extend our previous observations[27] that the nucleotide-loading status of the Gtrs dictates TOROID formation; whether generated by nutrient levels, mutations in the GAPs or mutations in the *GTR*s themselves, Gtr1[GTP]–Gtr2[GDP] antagonize whereas Gtr1[GDP]–Gtr2[GTP] promote TOROID formation. The EGOC also forms a punctum, which, in most but not all cases, correlates with the presence of a TOROID, the notable exceptions being the lack of EGOC puncta in Δ*gtr1* Δ*gtr2* cells and in exponentially growing cells expressing 'inactive' GTRs. These last results demonstrate that EGOC puncta are not necessary for TOROID formation but can assemble with TOROIDs in a Gtr-dependent manner. By contrast, active EGOC is needed to prevent TOROID formation.

We next performed in vitro binding assays between EGOC variants and free TORC1 dimers purified from exponentially growing cells or TOROIDs from PDS cells. The 'active' EGOC interacted preferentially, albeit weakly, with TORC1 dimers (Fig. 2h,i), while 'inactive' EGOC interacted robustly with TORC1 dimers and even more so with TOROIDs (Fig. 2h,i). Thus, in contrast to the situation in mammalian cells[21], the EGOC binds TORC1 regardless of which guanine nucleotides are bound to the Gtrs.

## High-resolution cryo-EM TOROID structure

To better understand the molecular regulation of TOROID (dis) assembly, we purified TORC1 from *KOG1-TAP Δtor1* cells grown PDS and acquired a high-resolution TOROID structure by cryo-EM. We hypothesized that TOROIDs containing exclusively Tor2 would yield a higher-resolution structure. Initial structure determination attempts using helical reconstruction resulted in three-dimensional (3D) reconstructions at rather low resolution (Fourier shell correlation $(FSC)_{0.143} = 9.1$ Å). The relatively poor precision of helical symmetry determination on the unsymmetrized map (Fig. 3a and Extended Data Figs. 3 and 4) suggested that instead of being a rigid helix, TOROID filaments resemble a flexible slinky-like spring with varying pitch. To overcome this issue, we employed signal subtraction protocols[43,44], using the known positions of TORC1 units in the helical reconstruction, to obtain a set of signal-subtracted particles that could be further analyzed using regular single particle analysis (SPA) (Extended Data Fig. 3).

Using these particles for 3D reconstruction (Methods)[45] resulted in a 3.87-Å map (FSC = 0.143), which enabled us to build a nearly complete

structural model of yeast TORC1 in the TOROID (Fig. 3b, Extended Data Fig. 4 and Table 1). Although the final map showed resolution heterogeneity (Extended Data Fig. 4a,b), the symmetric nature of the TOROID could be exploited to unambiguously build the majority of the TORC1 structure, with many regions resolving individual amino acid side chains (Fig. 3c). Moreover, the 3D reconstruction and local resolution estimation showed that the interactions between adjacent TORC1 units along the TOROID helix (intracoil) were more prominent than interactions along the helix *z* direction (intercoil; Fig. 3a and Extended Data Fig. 4a,b), consistent with the resolution difference at the intracoil and intercoil interfaces.

The overall yeast TORC1 structure extracted from TOROIDs was very similar to its mammalian counterpart (mTORC1; Fig. 4 and Extended Data Fig. 5). Whereas yeast and human Lst8 were virtually identical (r.m.s. deviation (r.m.s.d.) = 0.9 Å over 252 aligned main chain atoms), the main differences between Tor2/mTOR (r.m.s.d. = 3.3 over 1608 aligned main chain atoms) and Kog1/RAPTOR (r.m.s.d. = 2.4 over 866 aligned main chain atoms) could be traced to a conformational difference in Tor2 and the presence of previously uncharacterized additional α-helical regions for Kog1 (Fig. 4b,c). We compared our structure of yeast TORC1 present in TOROIDs with structures of mTORC1 with basal activity (nonactivated state) and active mTORC1 bound to Rheb[16]. Tor2 in TOROIDs is structurally more like mTOR in nonactivated-state mTORC1 ($r.m.s.d._{inactive}$ = 3.3 Å) than in Rheb-bound mTORC1 ($r.m.s.d._{active}$ = 10.4 Å; Fig. 4b). Indeed, TORC1 in TOROIDs had an even more 'open' conformation than nonactivated-state human mTORC1 (Fig. 4a): the HORN region of Tor2 made 10° and 24° outward shifts compared with nonactivated-state and Rheb-bound mTORC1, respectively (Fig. 4b).

Alignment of *Saccharomyces cerevisiae* Kog1 with orthologous sequences from *Homo sapiens, S. pombe* and *Chaetomium thermophilum* reveals a 'Twix' region contributed by two α-helices connected by a polyglutamine linker uniquely present in budding yeast (Fig. 4c,d and Extended Data Fig. 5c,d). The Twix region is situated between α-helices 26 and 29 of Kog1 and interacts with the FRB' domain of Tor2' in an adjacent TORC1' in the TOROID (Fig. 4d). Adjacent to the Twix region, we observed a map region corresponding to an α-helix not directly connected to surrounding parts of the structure, yet near Lys1016 of Kog1, which terminates α-helix 34 (residues 1,005–1,016, Extended Data Fig. 5c), and termed this part of TOROID the 'Tack' region. Careful inspection of the sequence succeeding Lys1016 and ensuing model building in Coot led us to propose this region to be a helix with sequence 'P$_{1069}$MRTSLAKLFQSLGFSES$_{1086}$' (Figs. 3c and 4d and Extended Data Fig. 5d) of Kog1. Sequence alignments demonstrated that this region precedes the sequence that aligns with the RAPTOR 'Claw' that was shown to interact with RagC in RAPTOR–Rag–Regulator reconstructions[22,23]. In this vicinity, the structure displays several prominent molecular interactions leading to a robust TORC1-TORC1' binding interface that we term the hub of interactions. This hub is centered around Trp2207, which sits between Arg884, Lys885 and Arg888 of Kog1, forming a prominent cation-pi interaction with Arg884 situated near the tip of the Tack α-helix and the base of the Twix region (Fig. 4d). Altogether, the interactions within this hub shield the kinase active site and thereby block substrate access, potentially contributing (see below), together with the presumed TORC1 subunit allosteric change described above, a structural explanation for why TORC1 is inactive in TOROIDs (Fig. 4d).

## Mutations confirm observed TORC1-TORC1' interfaces

To interrogate the functional importance of the intercoil and intracoil interfaces, we mutated selected residues using CRISPR–Cas9 (Fig. 5a,b and Extended Data Fig. 6) and assessed their impact on TORC1 puncta size in cells grown PDS (Extended Data Fig. 6e). In parallel, all mutants were tested in spot assays on rapamycin plates to assess TORC1 activity (Extended Data Fig. 6f). As controls, we first probed exposed residues that were in the vicinity of the intracoil and intercoil interfaces but

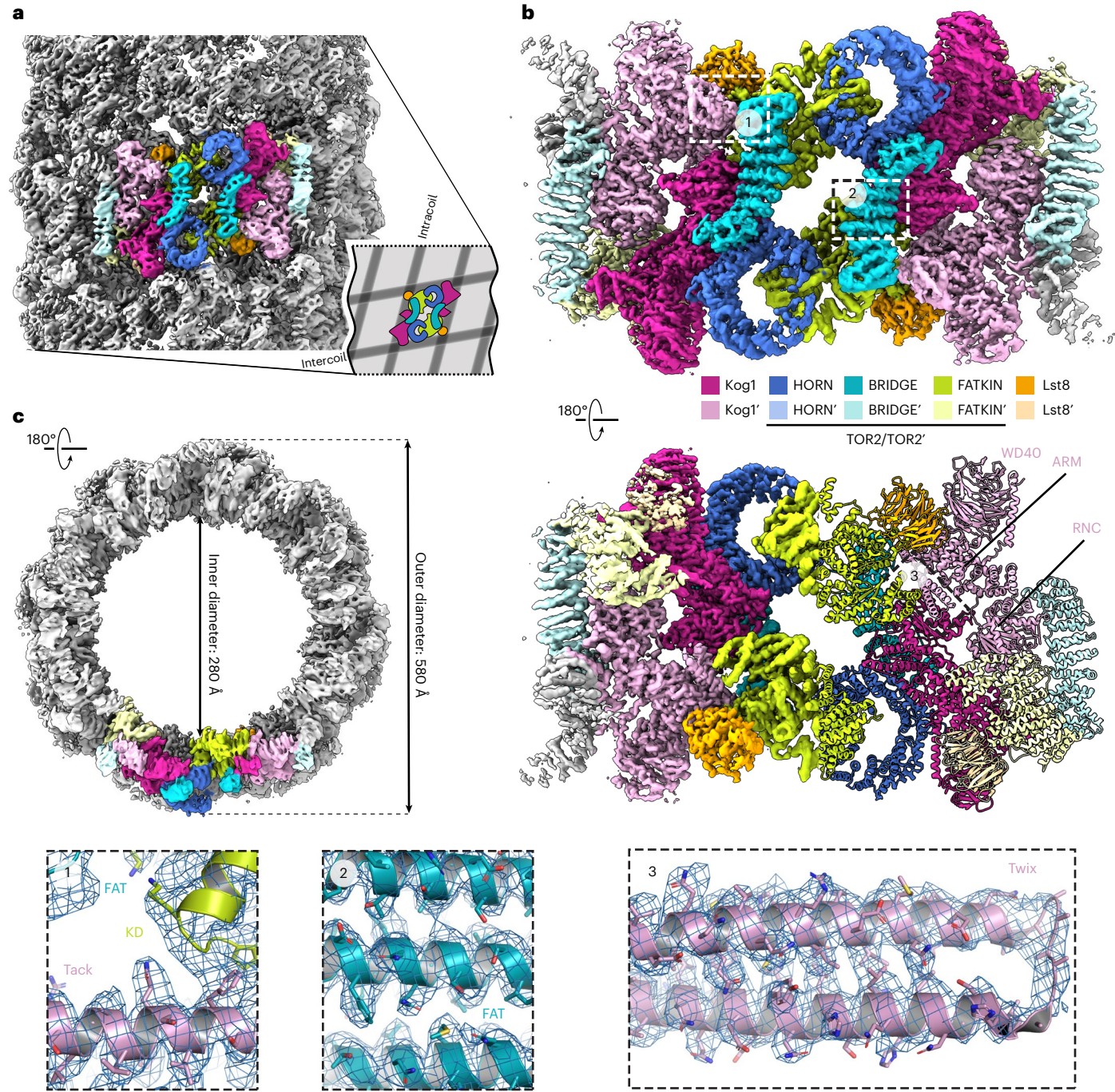

**Fig. 3 | TOROID Cryo-EM reconstruction. a**, Helical reconstruction TOROID map. **b**, Cryo-EM reconstruction of TOROID via signal subtraction and SPA. In **a**–**c**, coloring corresponds to different TORC1 subunits as indicated in **b**. **c**, Enlarged views from **b** showing a cartoon representation of the atomic TORC1 model, built in the Cryo-EM map (blue mesh). Selected amino acid residues are represented as sticks.

not obviously engaged in TORC1-TORC1' interactions. As expected, these mutations (Kog1[K1383A] and Kog1[Δ1544-end]) did not significantly affect TOROID size or rapamycin sensitivity (Extended Data Fig. 6a,c,f). By contrast, mutation of Tor1[W1279], which is not predicted to mediate contacts within the helix but is predicted to stabilize the BRIDGE domain, yielded a reduction in TOROID size as well as rapamycin hypersensitivity, consistent with TORC1 particle destabilization (Extended Data Fig. 6a,c,f).

Intercoil interactions, although visible after helical reconstruction, appeared to be almost completely absent in the signal-subtracted map and only became apparent at high contour threshold levels. The intercoil interface was predominantly formed between the Tor2

HORN and the Kog1' WD domain (Fig. 3 and Extended Data Fig. 6b). Two exposed loops in the HORN, Tor2[337–345] (Tor1[326–334]) and Tor2[379–381] (Tor[368–370]), were substituted by poly alanine-glycine stretches. These strains displayed a significant reduction in TOROID size as well as an increase in rapamycin resistance (Extended Data Fig. 6b,d–f). A protruding region of the Tor2 FAT/kinase domain (FATKIN) also appeared to interact with Kog1'; mutation of this region of Tor1 (Tor1[1449/54/56A]) resulted in similar phenotypes (Extended Data Fig. 6b,d–f).

Intracoil interactions were plentiful, and mainly contributed by Kog1 interacting with Kog1', Lst8', and the FATKIN' and BRIDGE' regions of Tor2' from an adjacent TORC1' protomer unit. Glu784 and Ser896 of

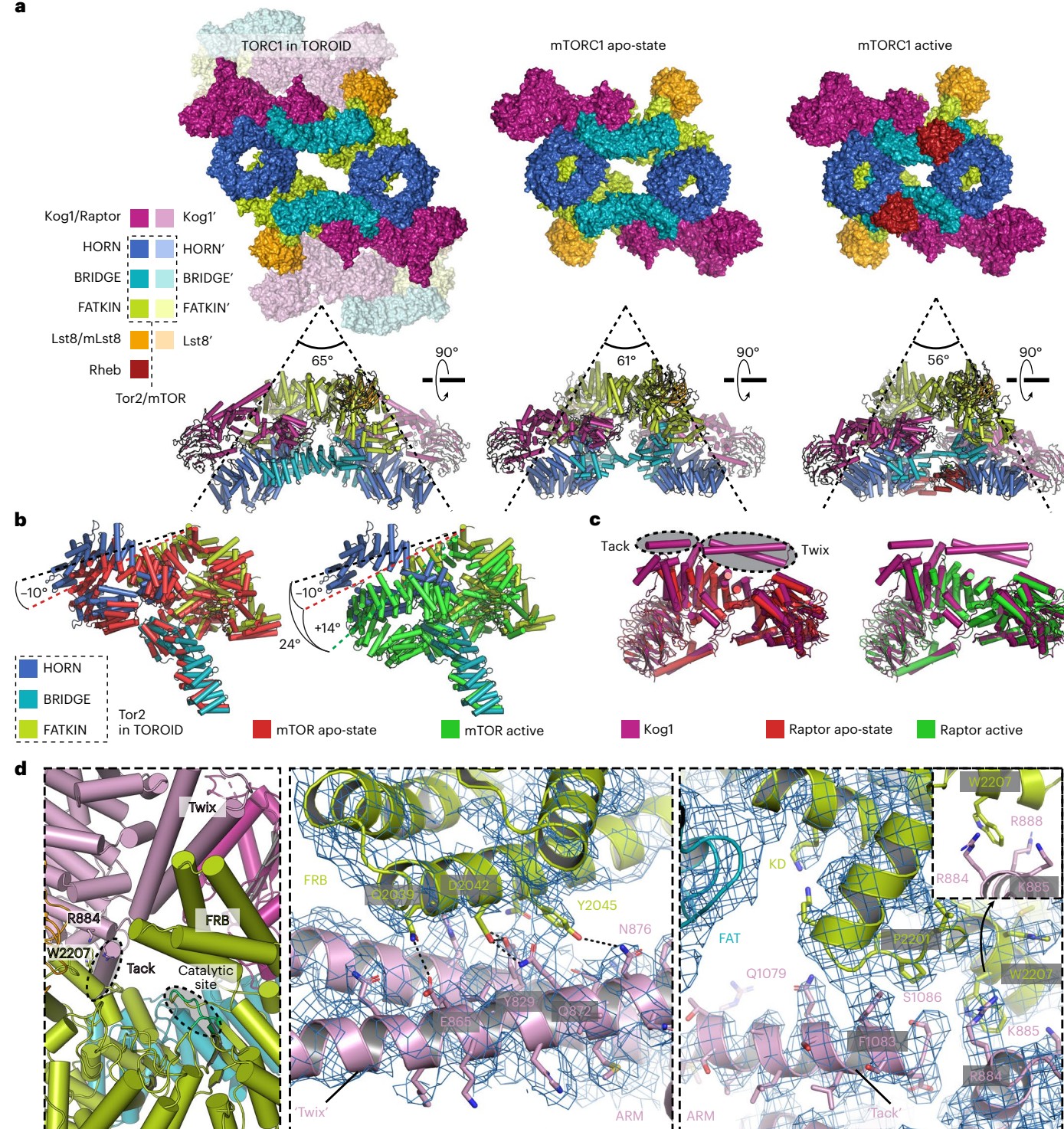

**Fig. 4 | TORC1 adopts an inactive conformation in TOROIDs. a**, Views of TORC1 in a TOROID (left), nonactivated mTORC1 (middle, PDB ID: 6BCX) and active mTORC1 bound to Rheb (right, PDB ID: 6BCU). In **a**–**d**, coloring of TORC1 subunits is performed according to the scheme in **a**. **b**, Structural alignment of Tor2 in TOROID with mTOR from nonactivated mTORC1 and mTOR from active mTORC1–Rheb. **c**, Structural alignment of Kog1 in TOROID with RAPTOR from nonactivated mTORC1 and RAPTOR from active mTORC1–Rheb. **d**, Inset showing an enlarged view of the hub of interactions between Kog1 and Tor2'. Enlarged views from **b** showing a cartoon representation of the atomic TORC1 model,

built in the cryo-EM map (blue mesh). The ATP-binding site (catalytic site) of the kinase domain is indicated. Selected amino acid residues are represented as sticks. The angles presented in **a** were obtained by making structural alignments in PyMOL and by directly measuring the angles on the rendered images. The angles presented in **b** were obtained using the 'get_angle' command in PyMOL, using atoms ASN'403/CA (mTOR in TOROID), ASP'1688/CA (mTOR in TOROID) and LYS'426/CA (mTOR inactive) for the first angle (9.8°) and atoms ASN'403/CA (mTOR in TOROID), ASP'1688/CA (mTOR in TOROID) and LYS'426/CA (mTOR active) for the second angle (23.9°).

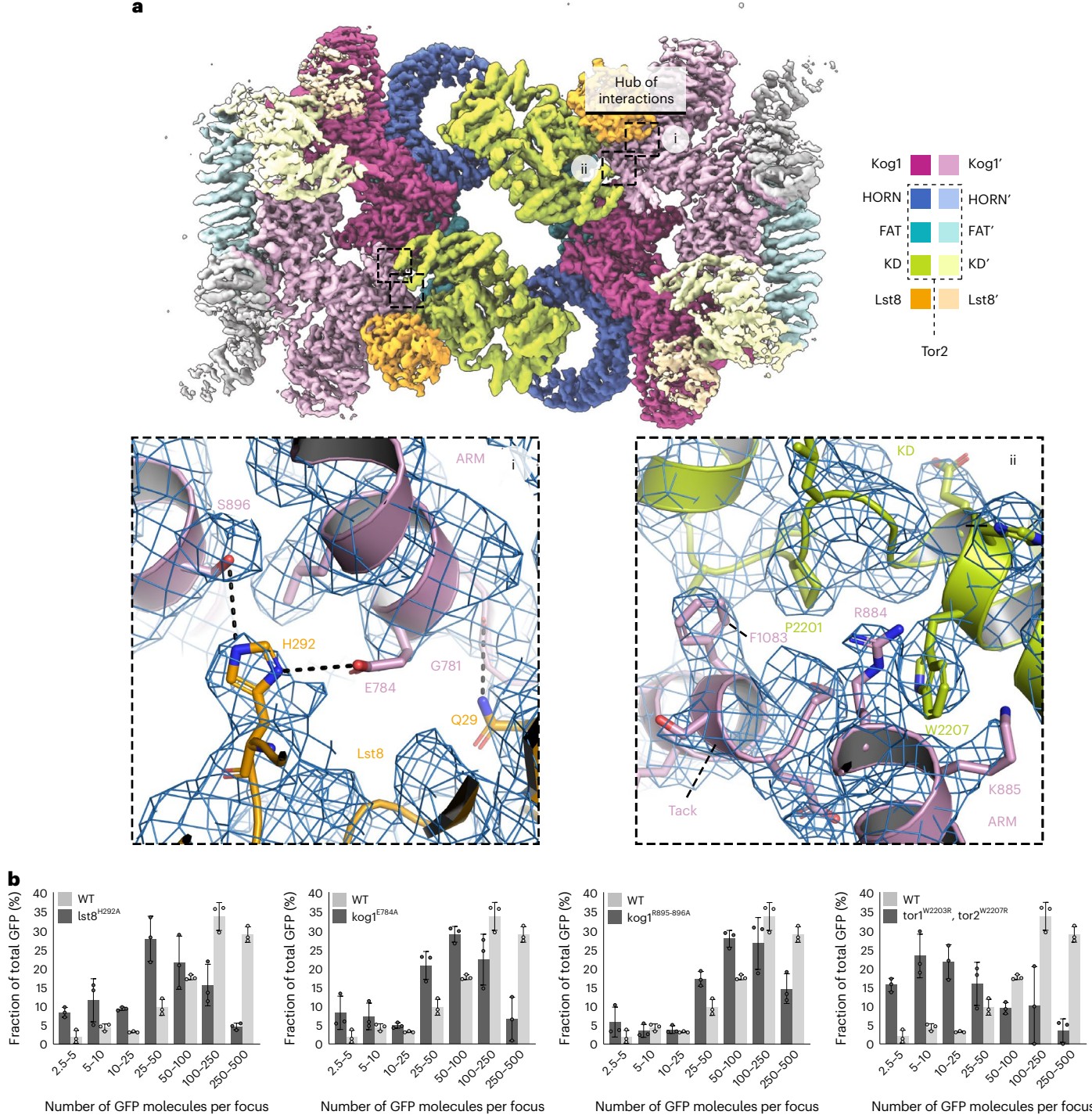

**Fig. 5 | CRISPR–Cas9 mutations to test intracoil and intercoil TORC1-TORC1' interactions. a**, TORC1 molecular model extracted from the TOROID cryo-EM map. Adjacent protomers are shown in light and dark colors. Dashed boxes (i, ii) indicate the regions mutagenized to validate the TORC1-TORC1' 'hub' of interactions. **b**, Distribution of GFP molecules in TORC1 foci measured in PDS cells harboring the indicated mutations. The data used to construct the graphs were generated based on analysis of at least 324 cells per replicate over $n = 3$ independent experiments. The control (WT) data are the same in all panels.

Kog1 interact with His292 of Lst8', and alanine substitution of this His reduced TOROID formation (Lst8^H292A; Fig. 5a,b). Similar results were obtained with reciprocal mutations in Kog1 (Kog1^E784A, Kog1^S895-896A; Fig. 5a,b and Extended Data Fig. 6e,f). Gly781 in Kog1 interacts with Gln29 of Lst8'; alanine substitution of this Gln residue strongly reduced TOROID formation and led to an increase in rapamycin resistance (Extended Data Fig. 6a,e,f).

Among the intracoil interactions, the TORC1-TORC1' 'hub' is formed through extensive contacts between the Twix helices of Kog1 and the FRB domain of Tor2' (Fig. 4d), the end of the Tack helix and the Tor2' FATKIN (Fig. 4d and Extended Data Figs. 5d and 6), and a basic pocket of Kog1 that hosts Trp2207 from the Lst8-binding element of Tor2'. The cation-pi interaction between Tor2^W2207 and Kog1^R884 was intriguing, as mutations in the corresponding residues of mTOR and

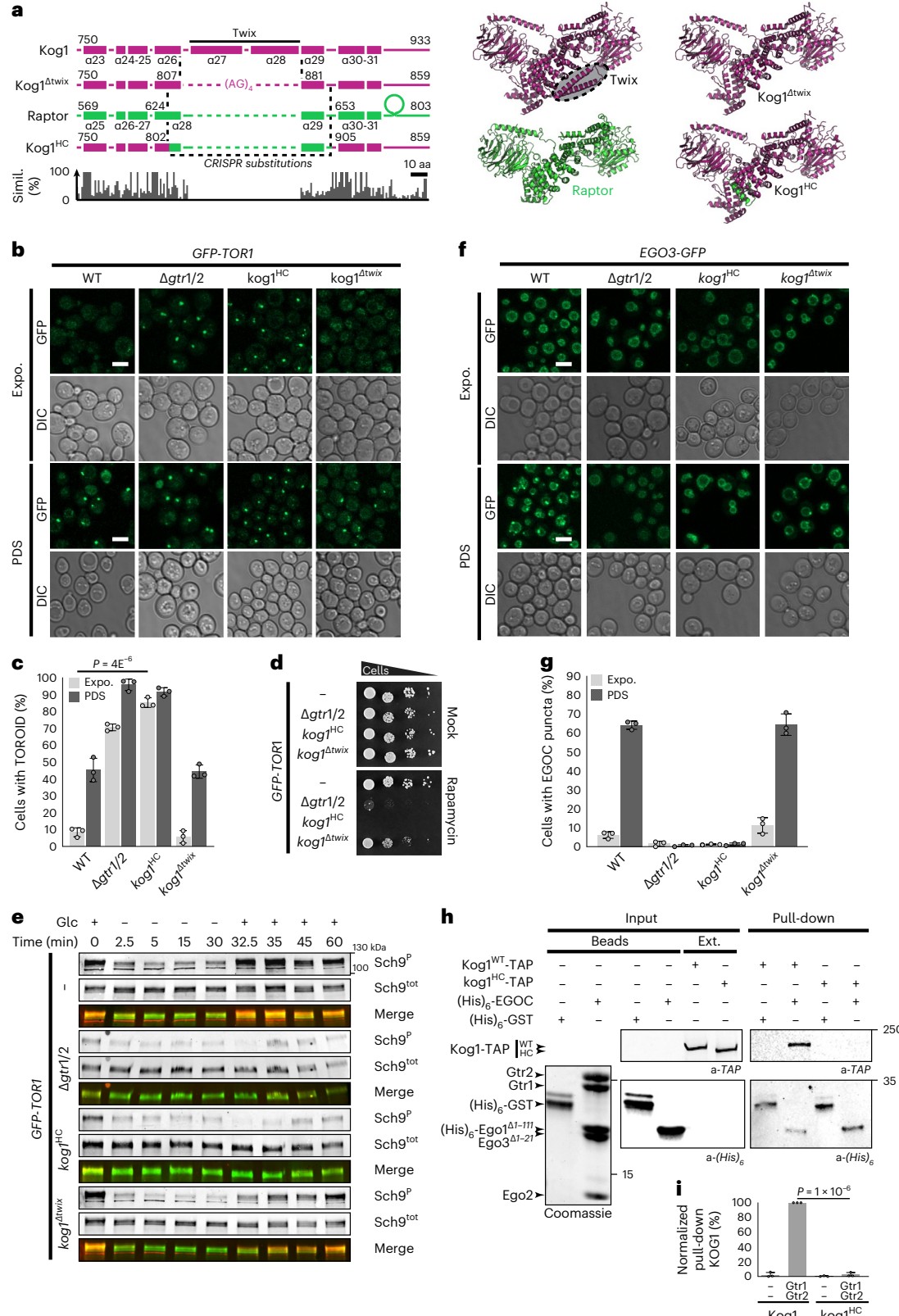

**Fig. 6 | Helices α26 and α29 of Kog1 are required for interaction with the EGOC. a**, Alignment of Kog1, Kog1[Δtwix], Raptor and Kog1[HC] sequences and corresponding structure models. Alpha helices are numbered according to the molecular model. **b,f**, WT, *Δgtr1 Δgtr2*, *kog1[HC]* and *kog1[Δtwix]* cells expressing *GFP-TOR1* (**b**) or *EGO3-GFP* (**f**) in exponential phase or in PDS (scale bars, 5 μm). **c**, Bar chart representing the percentage of cells with a TOROID from **b** based on *n* = 3 independent experiments with at least 101 cells examined. **d**, Cells with the indicated genotypes spotted onto plates containing 2.5 nM rapamycin.

**e**, Western blot analysis of glucose-dependent TORC1 activity using the indicated strains. **g**, Bar chart representing the percentage of cells with EGOC puncta from **f** based on *n* = 3 independent experiments with at least 69 cells examined. **h**, In vitro pull-downs of TORC1 containing either *KOG1-TAP* or Kog1[HC]-TAP using cobalt-Dynabeads precoated with purified (His)₆ proteins (Coomassie). Inputs and pull-downs were analyzed by western blotting. **I**, Quantification of **h** with *n* = 3 independent experiments. aa, amino acid; Glc, glucose; Simil., similarity.

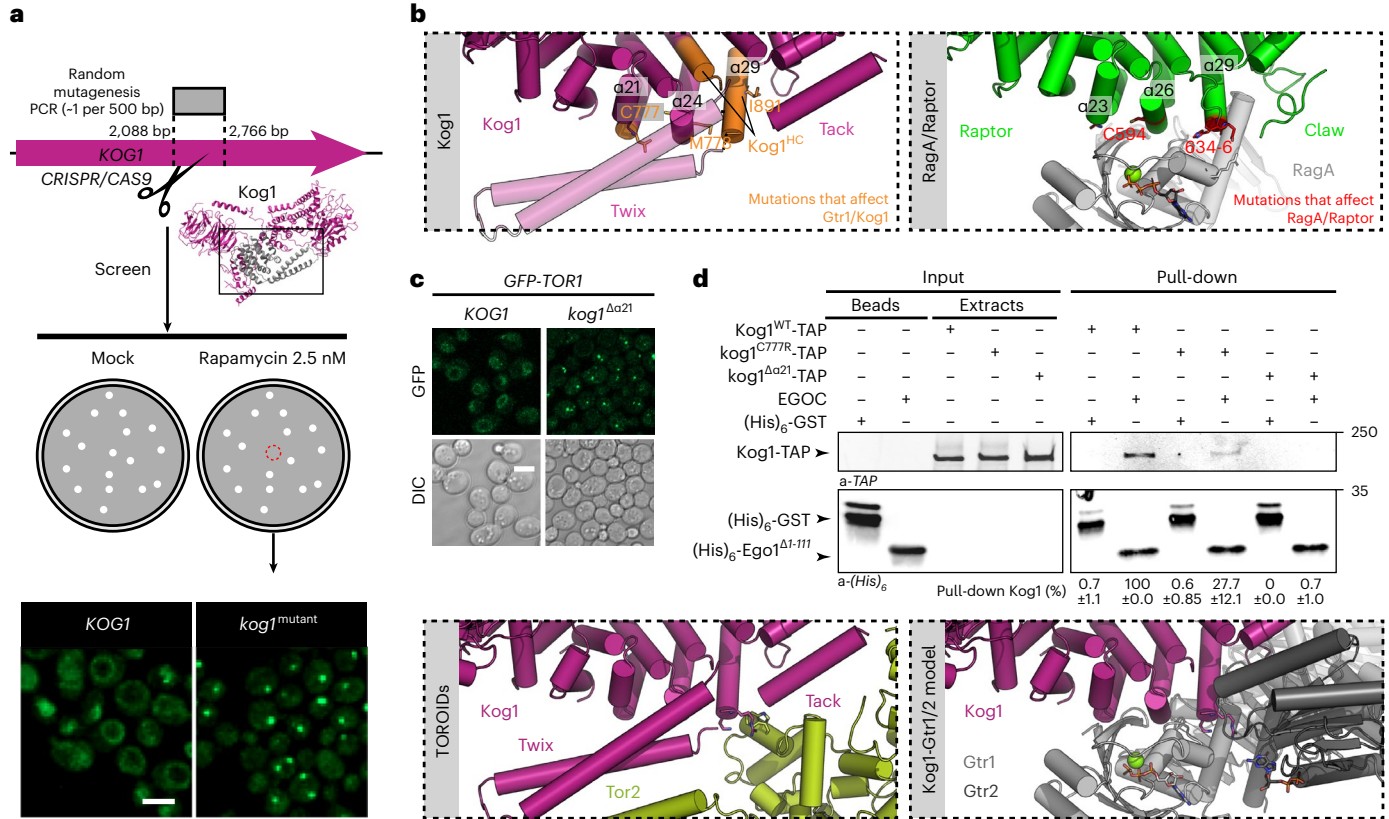

**Fig. 7 | Random mutagenesis reveals that EGOC binding to Kog1 competes with TORC1 oligomerization. a**, Schematic representing the random mutagenesis screen of the *KOG1armadillo* sequence to uncover constitutive TOROID mutants (scale bar, 5 μm). **b**, Left panel: locations of amino acids in Kog1 that when altered affected Gtr binding. Right panel: locations of amino acids in Raptor that when altered disrupted RagA binding[23]. **c**, Effect of Kog1 helix α21 deletion (*kog1Δα21*) on TORC1 (GFP-Tor1) localization in exponential-phase cells (scale bar, 5 μm). **d**, In vitro pull-downs of TORC1 containing either Kog1WT-TAP

or Kog1C777R-TAP or Kog1Δα21-TAP using cobalt-Dynabeads precoated with purified (His)₆ proteins. Inputs and pull-downs were analyzed by western blotting using the corresponding antibodies. **e**, EGOC and Tor1/2 compete for the same region within Kog1. Left panel: view of Kog1 interacting with Tor2′ as determined from our TOROID structure. Right panel: view of Kog1 interacting with Gtr1 and Gtr2 modeled from the Raptor–RagA–C structure and supported by our extensive mutagenesis experiments. bp, base pairs.

---

RAPTOR (mTOR^R2266P and Raptor^D635N) have been identified in human carcinomas (https://cancer.sanger.ac.uk/cosmic). To test if this interaction was important for TORC1 regulation, we substituted this tryptophan with arginine in both Tor1 and Tor2 (Tor1^W2203R, Tor2^W2207R). In these cells, TOROID formation was absent, and TORC1 was hyperactive as shown by increased rapamycin resistance, consistent with a charge repulsion with Kog1′ (Fig. 5 and Extended Data Fig. 6e,f). Substitution of Arg884 of Kog1 to Asp (the corresponding human residue; *KOG1^R884D*) partially blocked TOROID formation in PDS (30% reduction versus WT; Extended Data Fig. 7a,b). Multiple efforts to change codons R884/R885/K888 to alanine or aspartate unfortunately failed.

## *kog1^HC* allele unveils the TORC1-EGOC interaction domain

TOROIDs have not been reported in other systems, and we wondered whether the yeast-specific Twix helices α27 and α28 of Kog1 might be a reason for this. To test this, we made two *kog1* mutant strains, one (*kog1^ΔTwix*) in which the Twix helices α27 and α28 were 'cleanly' replaced with a short (AG)₄ linker and a second, (*kog1^HC*, humanized chimera) in which α26 to α29 were replaced with the corresponding, shorter sequence of human Raptor (Fig. 6a). *kog1^ΔTwix* cells formed TOROIDs normally when cultured PDS (Fig. 6b,c), had nearly normal sensitivity to rapamycin (Fig. 6d) and displayed essentially WT Sch9 phosphorylation kinetics upon acute glucose starvation and repletion (Fig. 6e). Kog1^HC cells cultured PDS also formed TOROIDs

indistinguishably from WT cells (Fig. 6b and Extended Data Fig. 6e). Thus, the Twix helices are not necessary for TOROID formation or TORC1 regulation.

The *kog1^HC* allele replaces helix α29, which contains the three basic residues (Kog1^R884/K885/R888) thought to interact with Trp2207 of Tor2 (Extended Data Fig. 5c). Although *KOG1^R884D* did not suppress the inability of *TOR1^W2203R TOR2^W2207R* cells to form TOROIDs, we found that *kog1^HC* did do so (Extended Data Fig. 6e). As expected, the *TOR1^W2203R TOR2^W2207R kog1^HC* strain had lower rapamycin resistance than the *TOR1^W2203R TOR2^W2207R* strain (Extended Data Fig. 6f). However, we unexpectedly noticed that *kog1^HC* cells contained TOROIDs when growing exponentially (Fig. 6b,c) and displayed hypersensitivity to rapamycin (Fig. 6d), like Δ*gtr1* Δ*gtr2* cells. Furthermore, in *kog1^HC* cells, TORC1 activity was reduced and largely insensitive to acute changes in glucose levels (Fig. 6e), and EGOC puncta were absent (Fig. 6f,g). Last, unlike WT TORC1, Kog1^HC-containing TORC1 failed to robustly interact with purified EGOC (Fig. 6h,i and Extended Data Fig. 7c). Collectively, these results suggest that the EGOC-binding site lies within helix α26 and/or α29 of Kog1, that is, within the TOROID hub of interactions. The results also demonstrate that TOROID assembly is necessary but not sufficient for EGOC puncta formation.

Based on these observations, we wondered whether *kog1^HC* suppressed the inability of *TOR1^W2203R TOR2^W2207R* cells to form TOROIDs because of its own inability to bind the EGOC rather than, or in addition

**Table 1 | Cryo-EM data collection, refinement and validation statistics**

| | TOROID (helical reconstruction, EMDB-13595) | TOROID (signal subtraction and SPA, EMDB-13594, PDB 7PQH) |
|---|---|---|
| **Data collection and processing** | | |
| Magnification | 37,000 | 37,000 |
| Voltage (kV) | 300 | 300 |
| Electron exposure (e⁻/Å²) | 20 | 20 |
| Defocus range (µm) | 1.0–2.5 | 1.0–2.5 |
| Pixel size (Å) | 1.35 | 1.35 |
| Symmetry imposed | D1 | D1 |
| Initial particle images (no.) | 219,455 | 219,455 |
| Final particle images (no.) | 219,455 | 218,872 |
| Map resolution (Å) | 9.10 | 3.87 |
| FSC threshold | 0.143 | 0.143 |
| Map resolution range (Å) | 11.5 – 7.5 | 6.0 – 3.6 |
| **Refinement** | | |
| Initial model used (PDB code) | NA | N.A. |
| Model resolution (Å) | NA | 4.30 |
| FSC threshold | | 0.5 |
| Model resolution range (Å) | NA | ∞ – 4.30 |
| Map sharpening $B$ factor (Å²) | −979.5 | −131.8 |
| Model composition | NA | |
| Non-hydrogen atoms | | 102,675 |
| Protein residues | | 12,832 |
| Ligands | | 0 |
| $B$ factors (Å²) | NA | |
| Protein | | 127.1 |
| Ligand | | – |
| R.m.s. deviations | NA | |
| Bond lengths (Å) | | 0.005 |
| Bond angles (°) | | 1.197 |
| Validation | NA | |
| MolProbity score | | 1.88 |
| Clashscore | | 8.74 |
| Poor rotamers (%) | | 0.12 |
| Ramachandran plot | NA | |
| Favored (%) | | 93.81 |
| Allowed (%) | | 6.16 |
| Disallowed (%) | | 0.03 |
| Ramachandran $Z$ score | NA | |
| Whole | | −1.40 |
| Helix | | −0.19 |
| Sheet | | −0.53 |
| Loop | | −2.01 |

to, charge complementation. Consistent with this idea, we observed that the Δ*gtr1* Δ*gtr2* double deletion was epistatic to *TOR1^W2203R TOR2^W2207R*: Δ*gtr1* Δ*gtr2 TOR1^W2203R TOR2^W2207R* cells presented constitutive TOROIDs (Extended Data Fig. 7d).

**Exclusive interaction of TORC1 with EGOC or TORC1′.** We observed a strong anticorrelation between TOROID size in PDS and rapamycin sensitivity among our mutant strains (PC: −0.75; Extended Data Fig. 7e). Strains with mutations compromising TORC1-TORC1′ interactions have small TOROIDs and are resistant to rapamycin, whereas those with mutations abrogating TORC1-EGOC binding have large TOROIDs and are sensitive to rapamycin. Thus, sensitivity to rapamycin depends on the proportion of free and active TORC1 dimers. Exploiting this idea, we set up a CRISPR-based PCR random mutagenesis screen, targeting the region surrounding α26–α29 (Kog1^696–922), to define further the TORC1-EGOC interface (Fig. 7a). The first selection step was rapamycin hypersensitivity, expecting that mutants of interest would present a hypersensitive phenotype like that of Δ*gtr1/2* and *kog1^HC* strains. The second step was to select for TOROID formation in glucose-replete cultures. From a library of 3,500 mutants, 42 rapamycin-sensitive clones were identified, of which 13 presented TOROIDs in glucose-replete media (Fig. 7a and Extended Data Fig. 8a,d). Seventeen point mutations were identified, of which seven (marked in bold below) were the only mutation present in the clone and could thus be deemed causative (Fig. 7b and Extended Data Fig. 8b–d). Remarkably, although the Twix-encoding region was included in this screen, no mutation within this sequence was recovered, supporting the idea that it does not participate in EGOC-dependent regulation of TOROIDs.

Several mutations, alone (in bold) or in combination, altered leucine residues involved in hydrophobic interactions between α-helices (**L762P**, L765P, **L766P**, **L900P**, **L912Q**; Fig. 7b and Extended Data Fig. 8c,d), suggesting that local perturbation of domain packing may affect EGOC binding. L737Q (found in combination with L762P), at the junction of the α21-22 helices, **C777R** and M778T in helix α24, **I802N** and **A804E** in helix α26, and I891N in helix α29 were identified (Fig. 7b and Extended Data Fig. 7b–d) and mapped to a region which in Raptor is important for interaction with RagA (Raptor residues N557, C594 and 634–636; Fig. 7b and Extended Data Fig. 5c)[23]. Consistently, replacing Kog1 helix α21 (corresponding to α23 in Raptor, which is important for RagA binding) with an alanine-glycine linker resulted in constitutive TOROID formation (Fig. 7c). We next queried whether 'activity' of the EGOC was important for TORC1-TOROID binding. Strikingly, we found that whereas TORC1^WT interacted with both active and inactive EGOC, TORC1 harboring Kog1^HC, Kog1^C777R or Kog1^Δα21 lost the ability to interact with active EGOC but retained the ability to interact with inactive EGOC (Fig. 7d and Extended Data Fig. 8e), suggesting that there are multiple binding modes between EGOC and TORC1.

In TOROIDs, the Tack helix is at the TORC1-TORC1′ interface and occupies a location similar to that of the Claw fragment of Raptor that interacts with RagC. To probe the function of Tack, we made several polyAG substitutions. Replacement of the proximal α-helix 34 (Kog1^1004–1022) had no effect on TOROID regulation, whereas replacement of residues 1,069–1,086 of Kog1, which we attribute to the TOROID Tack helix, caused a mild but significant impairment of TOROID formation (Extended Data Fig. 8f,g). Replacement of the putative yeast Claw-equivalent sequence (Kog1^1121–1131)[23] had no significant effect on either TOROID formation or regulation (Extended Data Fig. 8f,g). Thus, the Tack helix does not function equivalently to the Raptor Claw but provides structural elements (Kog1^F1083 binding to Tor2^′P2201) needed to stabilize the TOROID.

Based on these collective observations, we propose that, analogous to the binding observed in mammalian cells, active EGOC binds to helices α21, α24, α26 and α29 of Kog1. This binding competes with a critical interface needed for TOROID formation and would thus release free TORC1 dimers that are competent to signal to downstream effectors (Fig. 7e). Inactive EGOC appears to also bind via an independent, yet to be determined, interface within TORC1-TOROIDs.

## Discussion

We previously reported[27] that glucose withdrawal triggers a rapid, Gtr1Gtr−2-dependent reorganization of TORC1 from a disperse

localization over the vacuole surface to a hollow helical assembly named a TOROID. Our previous, low-resolution, 3D reconstruction revealed that TOROID-sequestered TORC1 is inactive owing to a rapamycin-FKBP12-like occlusion of the kinase active site.

We show here that the EGOC alone is sufficient to control the dynamic equilibrium between active TORC1 dimers diffusely localized on the surface of the vacuole, and inactive TORC1 dimers polymerized into a TOROID. Consistently, across many mutant strains, we observed a strong anticorrelation between TORC1 puncta size and rapamycin resistance (Extended Data Fig. 7c). We note that the EGOC is not the sole regulator of TORC1 activity (indeed TORC1-mTORC1 activity is still apparent in *gtr1 gtr2/rag* mutants[46]). How regulators such as Snf1 and Pib2 (refs. [47,48]) interface with TOROID-based regulation remains an important unknown. Furthermore, distinct pools of TORC1 (for example, endosomal TORC1)[26] could be subject to distinct upstream regulatory mechanisms.

The major advance of our present work is the molecular insight into how EGOC binding liberates TORC1 dimers from TOROID polymers. Our high-resolution TORC1 map revealed that TOROIDs are rather spring-like with loose intercoil interactions and robust intracoil interactions. The intracoil interactions occlude access to the active site, which, together with presumed allosteric changes to the catalytic spine, ensures inactivation of TORC1 engaged in a TOROID. Closer inspection of the TORC1-TORC1′ interface proximal to the kinase active site revealed a hub of interactions primarily involving Kog1 and Tor2′. Using CRISPR–Cas9-supported random mutagenesis of this region, we found that α-helices 21–26 and 29 of Kog1 serve as a binding platform for Gtr1–Gtr2. This observation was satisfying because the corresponding region of Raptor (α23–α29) had previously been found to bind to RagA–RagC[23].

Collectively, these observations clearly demonstrate that EGOC binding to this region of Kog1 disrupts the hub of interactions needed to sustain TOROID assembly, providing a mechanism to explain how EGOC can liberate and/or maintain free, active TORC1 dimers. This conclusion was not obvious because the positioning of the Twix helices should occlude this mode of GTPase binding.

Indeed, the Twix (Kog1 α27 and α28) and Tack (proposed residues 1,069–1,086 of Kog1) helices formed prominent features near the active site of an adjacent TORC1 in the TOROID structure. Modeling of the Twix helices could be unambiguously performed due to their direct connection with neighboring Kog1 residues. Conversely, the Tack region corresponds to an isolated helix not directly connected to Kog1, yet in close proximity to Kog1 Lys1016 (Extended Data Fig. 5c). The medium local resolution (Extended Data Fig. 5d) allowed us to hypothesize that the Tack helix corresponds to Kog1 residues 1,069-1,086. We argued from their placement that the Twix and Tack helices could be important in TOROID stabilization and TORC1 regulation. However, this appears not to be the case. Removal of the Twix helices slightly increased TOROID formation and rapamycin sensitivity, suggesting that they promote or stabilize active TORC1. By contrast, removal of the Tack partially compromised TOROID formation, suggesting that it helps to stabilize the TOROID structure. We hypothesize that these helices are mobile and are displaced upon EGOC binding. The molecular details of this displacement await a structure of TORC1 bound to active EGOC.

Although this model of TORC1 regulation is appealing, it is oversimplified, as it does not consider the nucleotide-loading status of Gtr1 and Gtr2. A priori, one could posit that only 'active' EGOC binds TORC1, preventing its association into a TOROID. However, the following compelling evidence suggests that inactive EGOC also binds TORC1.

(1) In in vitro pulldowns, TORC1 interacted with both 'active' and 'inactive' EGOC, apparently through distinct interfaces (Figs. 2h,i and 7d and Extended Data Fig. 8e).
(2) 'Inactive' EGOC puncta partially colocalize with TOROIDs (Supplementary Fig. 2).
(3) Expression of 'inactive' *gtr* alleles produce phenotypes that are more severe than those resulting from simple *gtr1/2* deletion[25].

Consistently, the expected accumulation of Gtr1^GDP/–Gtr2^GTP appears to cause synthetic lethality when deletion of *LST4* or *LST7* is combined with hypomorphic *sec13* alleles (encoding a SEACAT component)[49].

(4) Loss of RAG GTPases in *S. pombe* leads to hyperactivation of TORC1 (ref. [50]).

The RAG GTPases do not solely interact with TORC1; they also interact with their respective GAPs. Recent evidence suggests that with both GATOR1 and Folliculin-Fnip, the RAGs also have multiple binding modes[18,51,52]. How inactive EGOC inhibits TORC1 activity will be an important question for future studies.

Considering TOROIDs beyond yeast, there does not appear to be any structural argument to preclude the existence of TOROID in mammalian cells. Indeed, mTOROID-like regulation could explain why mTOR^R2266P and possibly Raptor^D635N are associated with carcinoma, based on our finding that substitution of the analogous residue in Tor1^W2203R Tor2^W2207R generated cells that did not form TOROIDs and appeared to possess hyperactive TORC1 signaling (that is, were resistant to rapamycin). This double mutant is intriguing as it still requires the Gtrs to prevent TOROID formation. It may possess a higher affinity for low levels of 'active' EGOC in PDS cells. Further characterization of this mutant is clearly warranted.

## Online content

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

## Methods

### Strains, plasmids, oligonucleotides and media

Yeast strains and plasmids are listed in Supplementary Tables 1–3. Yeasts were grown in complete synthetic medium (CSM) buffered at pH 6.25 (Sorensen Buffer), lacking the appropriate amino acid(s) to maintain plasmid selection when necessary.

### Yeast growth conditions

Saturated yeast liquid precultures grown in CSM were diluted into fresh CSM at optical density measured at a wavelength of 600 nm ($OD_{600}$) of 0.3 and cultured for 4 h at 30 °C to reach exponential growth ($OD_{600}$ ~0.6) or for 24 h to reach PDS. Confocal Z-stack images were acquired on an LSM800 confocal microscope (Plan-Apochromat 63x/1.4 Oil Objective, Zeiss) at the relevant growth states.

### Growth assays

Exponentially growing cells normalized to $OD_{600}$ 0.1 (5 µl, from four serial ten-fold dilutions) were spotted on selective CSM plates containing either rapamycin (at the indicated dilutions), Mohr salt (100 µM) or BPS (25 µM) and grown at 30 °C for 2–3 days. To measure the half maximal effective concentration ($EC_{50}$) of rapamycin, plates were imaged and the intensity of each spot was measured using ImageJ. At each rapamycin concentration, a relative growth coefficient was measured as the slope of the linear regression based on the intensities of the four dilution spots. The relative growth coefficients were then plotted as a function of the rapamycin concentration and $EC_{50}$ values were obtained with Prism software (v.8.3.0).

### Iron deprivation treatment

$EGO1^{TEV}$ cells transformed with a p($CTH2p$)-3HA-TEV-(Cth2t) plasmid were grown overnight in buffered CSM lacking uracil (CSM-URA) supplemented with 0.3 µM Mohr salt (($NH_4$)$_2$Fe($SO_4$)$_2$). Cultures were diluted to $OD_{600}$ 0.2–0.3 and grown for 5 h before the iron deprivation treatment was started. This ensures the repression of the $CTH2$ promoter. Cells were filtered with hydrophilic mixed cellulose ester membranes of 65 µm pore size and rinsed with buffered CSM-URA to wash out potential aggregates of Mohr salt. Cells were resuspended in buffered CSM-URA supplemented with 0.1 µM BPS and either imaged by confocal microscopy or harvested for immunoblotting assays at different time points during 90 min.

### Glucose deprivation and repletion

At $OD_{600}$ 0.6, cells growing exponentially at 30 °C in liquid CSM were filtered, washed once with starvation medium (CSM-glucose) and then transferred to CSM-glucose. Glucose (2%, from a 20% stock solution) was subsequently added after 30 min of starvation. Confocal Z-stack images were acquired on an LSM800 confocal microscope and/or aliquots for western blot analyses were taken at the relevant time points.

### Yeast mating experiments

Experiments were performed by cospotting equivalent amounts of exponentially growing *MATa* and *MATalpha* cells onto a YPD plate and leaving them at 30 °C for 2.5 h. The yeast mixture was then subsequently resuspended in liquid CSM at 30 °C and deposited on VI 0.4 µ-Slides (ibidi) coated with concanavalin A (0.5 mg ml⁻¹) to be subsequently imaged by Z-stacks or time-series (1 image per 30 s) Z-stacks using a TCS SP5 confocal microscope (Plan-Apochromat 63×/1.4 Oil Objective, Leica).

### Light microscopy and analyses

All confocal images presented in the main and extended data are maximum projections of 3–5 images within a Z-stack (ImageJ). Z slices were initially separated by 500 nm. Image deconvolution was performed with Huygens software. Analysis of TORC1 foci into size or volume classes followed 3D reconstructions of the stack image, and intensity measurements were obtained using Imaris software[27]. The vacuolar enrichment of *EGO3-GFP*, *EGO1*[TEV] and control strains subjected to BPS treatment was measured with ImageJ. The mean intensity value in the cytosol was subtracted from the vacuolar membrane signal. The time required for TOROID formation or dissolution upon EGOC depletion or replenishment was analyzed with ImageJ by measuring the intensity of TORC1 focus across the Z-stacks and over the duration of acquisition. To quantify the degree of colocalization between TORC1 and EGOC, the intensity-based colocalization function of the Imaris software was used to extract the MC and PC. Alternatively, measurement of reciprocal puncta colocalization was performed using cell segmentation (FIJI) followed by segmentation of puncta according to their size in each channel (less than or greater than 500 nm; Imaris). Signals from both channels were extracted and subjected to Pearson analysis to measure their correlation.

### CRISPR–Cas9 mutageneses

A pML104 plasmid expressing Cas9 enzyme and guide RNA (from Addgene) was linearized by PCR (Key Resource Table), and the 20-mer guide sequence targeting a specific yeast genome site was inserted by Gibson cloning[53]. Mutagenesis was achieved by genomic cut and recombination performed by cotransforming the yeast strain with the specific plasmid and the codon-optimized PCR product[54]. Short mutagenic PCR products (<100 nucleotides) were obtained using overlapping forward and reverse primers (Supplementary Table 1), whereas long mutagenic PCR products (>100 nucleotides) were obtained using codon-optimized gene sequences (GenScript; see Key Resource Table). Yeasts were plated onto CSM-URA plates and then replicated on the same plates to decrease the number of false positive clones. CRISPR efficiency was estimated by comparing the number of clones obtained with and without cotransformation with PCR product. Ten clones were then subjected to colony PCR (and subsequently sequenced to verify the mutagenesis). CRISPR–Cas9-based random PCR mutagenesis was performed using a GeneMorphII Random Mutagenesis Kit (Agilent Technologies), aiming for 1–2 mutation(s) per 500 bp of PCR product. Transformants were plated onto 10-cm CSM-URA plates at a density of 100–200 clones per plate and then replicated onto CSM-URA with or without 2.5 nM rapamycin. Rapamycin-sensitive clones were then sequenced using the same approach as previously described.

### Western blot analysis

Yeast cultures were treated following standard TCA-urea extraction procedures. Protein lysates were separated by 4–20-gradient sodium dodecyl sulfate polyacrylamide gel electrophoresis (SDS–PAGE) and transferred to a nitrocellulose membrane using the iBlot system (ThermoFisher). The membrane was probed with primary antibodies overnight at 4 °C, before being washed and incubated with secondary antibodies for 45 min at room temperature. The membrane was developed using the Odyssey imaging system (LI-COR), and the results were quantified using ImageStudio Lite (LI-COR).

The primary antibodies used were rabbit polyclonal anti-Sch9, mouse monoclonal anti-P-Sch9[S758] (in-house at dilutions of 1:5,000 and 1:2,500, respectively), mouse monoclonal anti-HA (1:15,000, Sigma no. H9658), mouse monoclonal anti-polyhistidine antibody, clone HIS-1 (1:4,000, Sigma no. H1029-2ML) and rabbit polyclonal anti-TAP antibody (1:2,000, Open Biosystems no. CAB1001). The secondary antibodies used were donkey anti-rabbit, IRDye 800 (Li-Cor Biosciences) and donkey anti-mouse, IRDye 680 (Li-Cor Biosciences). All secondary antibodies were used at a dilution of 1:10,000.

### Plasmid design and protein purification

For the expression of all components of the EGOC, two vectors were designed: one for the expression of EGO-TC and the other for the dimer of GTPases. Coding sequences of *EGO2(Δ1-21)-EGO3-EGO1(Δ1-111:His₆)* were codon-optimized for bacterial expression and synthesized by

GenScript in pUC57. All open-reading frames (ORFs) were separated by an operon sequence (ttaactttaaaaaaaaaaaaacaggaggcaatatacat). The synthetized fragment was digested with NcoI and XhoI. The tricistronic ORF was cloned by T4 DNA ligation into a vector derived from pACYC (digested with NcoI and XhoI). The DNA sequences encoding full-length *GTR1* and *GTR2* were amplified from *S. cerevisiae* gDNA (TB50 background). The bicistronic ORFs, separated by an operon sequence (aggaggaaaaaaaa), were cloned by T4 DNA ligation into a pET-derived vector. Alternatively, plasmids carrying mutations that lock Gtr1 and Gtr2 in their active or inactive conformations (Q65L/S23L or S20L/Q66L, respectively) were used as templates for PCR[25].

GTPases or all EGOC components were (co)expressed in *Escherichia coli* (BL21 DE3), and transformed cells were grown in 2× YT media at 37 °C for 4 h, followed by overnight induction at 18 °C with 0.1 mM IPTG. Cells were harvested by centrifugation and resuspended in lysis buffer for Ni-NTA purification (50 mM Tris-Cl pH 7.4, 300 mM NaCl, 5% glycerol, 20 mM imidazole, 0.15% CHAPS, 1 mM MgCl$_2$, 1 µg ml$^{-1}$ DNase, 1 µg ml$^{-1}$ lysozyme, supplemented with protease inhibitors 1 mM PMSF and cOmplete Protease Inhibitor Cocktail) and lysed using an Emulsiflex system (AVESTIN). Total lysate was cleared by centrifugation at 26,890g for 45 min at 4 °C. The soluble fraction was subjected to affinity purification using a chelating HiTrap FF crude column (GE Healthcare) charged with Ni$^{2+}$ ions on an AKTA-HPLC explorer. Proteins were washed and eluted in lysis buffer containing 250 mM imidazole. The purest fractions were concentrated to about 10 mg ml$^{-1}$ (Amicon 50 kDa) and loaded on a Superdex GF200 Increase, equilibrated with storage buffer (50 mM MES-NaOH pH 6.0, 300 mM NaCl, 0,15% CHAPS, 1 mM MgCl$_2$, 0.5 mM DTT). Purified proteins were concentrated to 1–1.8 mg ml$^{-1}$.

## Protein sequence alignments

Protein sequence alignments were done using sequences from *S. cerevisiae*, *S. pombe*, *C. thermophilum* and *H. sapiens* and generated with ClustalX2. The Kog1 ortholog sequences were: *S. cerevisiae*, QHB09177, *S. pombe*, CAB08769; *C. thermophilum*, XP_006691974; and *H. sapiens*, Q8N122. The Lst8 ortholog sequences were: *S. cerevisiae*, QHB11374; *S. pombe*, BAA32427; *C. thermophilum*, XP_006696584; and *H. sapiens*, Q9BVC4. The TOR ortholog sequences were: *S. cerevisiae* Tor1, CAA52849 and *S. cerevisiae* Tor2, CAA50548; *S. pombe* Tor2, NP_595359 and *S. pombe* Tor1, NP_596275; *C. thermophilum*, XP_006695016; and *H. sapiens* mTOR, P42345.

## 6×HIS pull-down assay

Cells expressing *KOG1-TAP* or *kog1$^{HC}$-TAP* were cultured in CSM and harvested in either exponential phase or PDS (OD$_{600}$ 2.5 or OD$_{600}$ 16, respectively). Cell pellets were frozen and further lysed by grinding in a cold mortar and pestle under liquid nitrogen. Cell extracts were resuspended in lysis buffer (50 mM Tris pH 7.4, 150 mM NaCl, 10 mM MgCl$_2$, 0.1% NP-40, 1 mM PMSF, three tabs of cOmplete Protease Inhibitor Cocktail per 100 ml of lysis buffer). Cell lysates were cleared by centrifugation at 4,000g at 4 °C for 10 min. Total protein concentrations were measured using Bradford protein assay and normalized to 5 mg ml$^{-1}$. Dynabeads His-Tag Isolation & Pulldown (Novex) were washed once with binding buffer (50 mM Na-MES pH 6, 600 mM NaCl, 1.5 g l$^{-1}$ CHAPS) and incubated with 20 µg of purified proteins (HIS$_6$-GST or Ego2-Ego3-His$_6$-Ego1, Gtr1 and Gtr2) for 30 min at 4 °C. Beads were washed once with lysis buffer and resuspended in the initial bead volume. For pull-down assays, 15 mg of total proteins and 25 µl of coated Dynabeads were used per condition. The mixes were incubated for 60 min at 4 °C. The beads were washed ten times with 2 ml ice-cold lysis buffer and boiled with 2× SDS−PAGE sample buffer to elute the proteins, which were loaded onto 4−20% Mini-PROTEAN TGX Precast Protein Gels. For immunoblot analysis, rabbit polyclonal anti-TAP and mouse monoclonal anti-polyhistidine antibodies were used.

## Cryo-EM sample preparation

TORC1 filament samples were adsorbed for approximately 1 min onto Lacey carbon film grids (300 microMesh), blotted with Whatman filter paper according to the method described in ref. [55], and subsequently plunged in liquid ethane using a homemade plunging apparatus.

## Cryo-EM data collection

Cryo-EM images of TORC1 filaments were collected on a Titan Krios microscope (Thermo Scientific) at EMBL Heidelberg, Germany, operated at 300 kV and equipped with a K2 summit direct electron detector (Gatan) camera operated in counting mode. A total of 4,901 movies of 16 frames were collected, with a dose rate of 2.5 e$^{-}$/Å$^2$/s and a total exposure time of 8 s, corresponding to a total dose of 20 e$^{-}$/Å$^2$. Movies were collected at a magnification of 37,000×, corresponding to a calibrated pixel size of 1.35 Å per pixel at the specimen level.

## Cryo-EM data processing

Motion correction and dose weighting of the recorded movies were performed using MotionCor2 (ref. [56]), discarding the first frame. Initial contrast transfer function (CTF) estimation was performed on the aligned and dose-weighted summed frames using CTFFIND4 (ref. [57]). TORC1 filaments were picked manually using the e2helixboxer module in EMAN2 (ref. [58]) from a total of 3,864 micrographs (1,037 micrographs did not contain any TOROID filaments), resulting in 8,625 picked filaments. A total of 219,455 particles were extracted in RELION2.0 (ref. [59]) using the -helix option, with an extract size of 600 pixels (810 Å), an outer diameter of 620 Å and a helical rise of 26.75 Å. Following particle extraction, per-particle CTF correction was performed using Gctf[60].

Initially, helical reconstruction was performed in RELION2.0 using binned particles (2.7 Å per pixel), a previously acquired low-resolution map of TOROID[27] low-pass filtered to 30 Å, and corresponding previously determined helical parameters as input, while imposing D1 symmetry and allowing for a helical symmetry search to optimize the respective helical parameters. The result of this helical reconstruction converged to a map with a resolution of around 9.1 Å based on the 0.143 gold-standard FSC criterion[61]. The 3D classification and subsequent helical refinement in RELION2.0 did not result in an increase in resolution. Next, we imported nonbinned particles and the reconstructed map resampled to a voxel size of 1.35 Å into cryoSPARC 3.01 (ref. [62]) and performed helical reconstruction imposing D1 symmetry, while allowing a helical symmetry search starting from the helical parameters obtained from the RELION helical reconstruction. The resulting 3D reconstruction had a resolution of 9.3 Å, similar to the helical reconstruction in RELION. The broad peak of the mean squared error in a plot showing the estimated twist and rise (Supplementary Fig. 3a) indicated a relatively poor accuracy for determining the helical symmetry parameters on the unsymmetrized map of the last 3D refinement round. Indeed, similar scores were obtained for a twist ranging from approximately 46.9° to 47.7° and rise from approximately 25.2 Å to 28.2 Å, which would translate into pitch changes from 201 to 205 Å and 192 to 215 Å, respectively (given an average twist for variable rise and vice versa).

We thus argued that the rather low resolution obtained after helical reconstruction in RELION and cryoSPARC could be the consequence of a varying pitch along the helices, resulting in an averaging of the structure while implying one set of helical parameters. To test this, we used the 9.1 Å helical reconstruction map (resampled to a voxel size of 1.35 Å) to create a mask containing the whole TOROID filament segment except for one central TORC1 assembly and its interacting regions with adjacent TORC1 complexes to ensure that information on the TORC1-TORC1′ interfaces remained intact. We then performed signal subtraction in RELION2.0, using this mask and the structure and segment coordinates of the helical reconstruction, to generate a set of 219,455 subtracted particles containing one isolated TORC1 assembly per segment. Next, we employed the 'localrec' module in

Scipion[43,44] to crop and recenter the signal-subtracted particles in a smaller box of 300 pixels (405 Å), as well as assigning to each particle a refined defocus based on the helical geometry. SPA of the subtracted particles using 3D refinement in RELION2.0, while imposing D1 symmetry, resulted in a map with a resolution of 4.5 Å (FSC = 0.143). We then imported the particles from the RELION2.0 refinement into cryoSPARC 3.01. After two-dimensional classification and class selection, a set of 213,808 selected particles was used for nonuniform refinement[45] in cryoSPARC 3.01, employing a dynamic mask, imposing D1 symmetry, using the option to keep particles from the same helix in the same half-set, and allowing high-order aberration estimation and correction, which resulted in a final map with a resolution of 3.87 Å (FSC = 0.143). A final postprocessed and locally sharpened map was obtained using DeepEMhancer[63]. Local resolution estimation of the final 3D reconstruction was calculated in cryoSPARC 3.01. A summary of cryo-EM data collection parameters and image processing procedures can be found in the Supplementary Information (Supplementary Fig. 3 and Table 1).

### Model building and refinement

Initial homology models of Tor2 and Lst8 were generated using Phyre2 (ref. [64]), whereas a model of Kog1 was generated using ITasser[65]. Homology models of Tor2, Lst8 and Kog1 were first manually placed in the final 3D reconstruction, followed by rigid-body fitting in Chimera[66]. The rigid-body fitted models were subsequently subjected to a round of flexible fitting using Imodfit[67], followed by automatic molecular dynamics flexible fitting using NAMDINATOR[68]. The flexibly fitted structure was then refined using the Phenix software package[69] employing global minimization, local grid search, atomic displacement parameters (ADP) refinement, secondary structure and Ramachandran restraints, and noncrystallographic symmetry constraints. Initial refinement was followed by several cycles of extensive manual building in Coot[70], followed by additional rounds of refinement in Phenix using a nonbonded weight parameter of 200. A final refinement round was performed in Phenix using global minimization, local grid search, ADP refinement, secondary structure and Ramachandran restraints, noncrystallographic symmetry constraints and a nonbonded weight parameter of 300.

Alphafold[71,72] structural predictions for Kog1, Lst8 and Tor2 were released during preparation of this manuscript. Individual subunits within the built structural model of the TOROID assembly as well as the corresponding EM map were thoroughly compared with the Kog1, Lst8 and Tor2 Alphafold predictions. The Alphafold structural prediction for Lst8 was virtually identical to the Lst8 extracted from our TOROID structural model (r.m.s.d. = 0.9 Å over 259 aligned main chain atoms). The Alphafold Tor2 model was predicted equally well but displayed a closed conformation (similar to active mTOR, Fig. 4b), in stark contrast to the open conformation of Tor2 extracted from our TOROID structural model, resulting in a r.m.s.d. value of 8.5 Å (1894 aligned main chain atoms). Whereas the Kog1 Alphafold model showed high overall agreement with the Kog1 extracted from the TOROID structure (r.m.s.d. = 0.8 Å over 1,016 aligned main chain atoms), the 'Twix' region displayed a tilt that markedly differed from the TOROID structure. Residues 1,069–1,086 Kog1 which we assigned to the 'Tack' region, although correctly predicted as a helix surrounded by long loop regions, is predicted to be in a different location. Accordingly, the model confidence score of this particular region in the Kog1 Alphafold structural prediction was very low. Nonetheless, we were able to use the Kog1 Alphafold model to improve the TOROID Kog1 structure in regions where the EM map was very weak, notably the Kog1 N terminus and the C-terminal WD domain, resulting in significantly improved protein geometry validation statistics. A summary of refinement and validation statistics can be found in Table 1.

### Statistics and reproducibility

Statistical significance was determined by three or more independent biological replicates unless stated otherwise in the legends. All data presented in plots were generated and analyzed using GraphPad Prism 9.4.1. Data are represented as mean ± s.d. with each dot representing the mean value of one replicate. Unpaired Student's $t$ test was used for comparisons of datasets. Measures with $P$ value lower than 0.05 were considered significantly different. Effect sizes and degrees of freedom can be extracted from the initial GraphPad Prism files (.pzfx) provided as raw data.

### Reporting summary

Further information on research design is available in the Nature Portfolio Reporting Summary linked to this article.

### Data availability

Cryo-EM TOROID maps obtained using either helical reconstruction or signal subtraction and ensuing SPA have been submitted to the EM Data Bank with accession codes EMDB-13595 and EMDB-13594, respectively. The atomic model built in the signal-subtracted TOROID cryo-EM map has been submitted to the Protein Data Bank with accession code 7PQH. Source data are provided with this paper.

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

## Acknowledgements

R.L. acknowledges support from the Canton of Geneva, the Swiss National Science Foundation, SNF Sinergia METEORIC, the Swiss National Centre for Competence in Research Chemical Biology and the European Research Council AdG TENDO. I.G. is supported by the European Union's Horizon 2020 research and innovation program under grant agreement no. 647784 to IG. S.N.S. is supported by the Flanders Institute for Biotechnology (VIB). J.F. was supported by a long-term European Molecular Biology Organization fellowship (ALTF441-2017) and a Marie Skłodowska-Curie Actions individual fellowship (789385, RespViRALI). Support and use of iNEXT resources and the cryo-EM service platform at EMBL Heidelberg and CryoGEnic facility (DCI Geneva) are acknowledged. J.F. and A.D. acknowledge the use of the Grenoble IBS-EMBL computer cluster and the technical assistance of Aymeric Peuch. This work used the EM facilities at the Grenoble Instruct-ERIC Center (UAR 3518 CNRS CEA-UGA-EMBL) with support from the French Infrastructure for Integrated Structural Biology (ANR-10-INBS-05-02) and GRAL, a project of the University Grenoble Alpes graduate school (Ecoles Universitaires de Recherche) CBH-EUR-GS (ANR-17-EURE-0003) within the Grenoble Partnership for Structural Biology. The IBS Electron Microscope facility is supported by the Auvergne Rhône-Alpes Region, the Fonds Feder, the Fondation pour la Recherche Médicale and GIS-IBiSA.

## Author contributions

Genetic engineering and cellular biology experiments were carried out by M.P., C.B., L.B. and R.L. EGOC expression and purification were performed by C.G., and TOROID purifications, pull-downs and cryo-EM grid preparations were performed by M.P. and C.B. EM data collection was performed by M.P., C.B., J.F. and A.D. Cryo-EM data processing, model building and refinement were carried out by J.F. and A.D., with additional input from Y.S. Experimental design was conducted by M.P., C.B., J.F., I.G., A.D. and R.L. The first draft of the manuscript was written by M.P., C.B., J.F. and R.L. with input from M.P., C.B., J.F., I.G., A.D. S.N.S and R.L.

## Funding

## Competing interests

The authors declare no competing interests.

## Additional information

**Extended data** is available for this paper at https://doi.org/10.1038/s41594-022-00912-6.

**Correspondence and requests for materials** should be addressed to Manoël Prouteau, Jan Felix or Robbie Loewith.

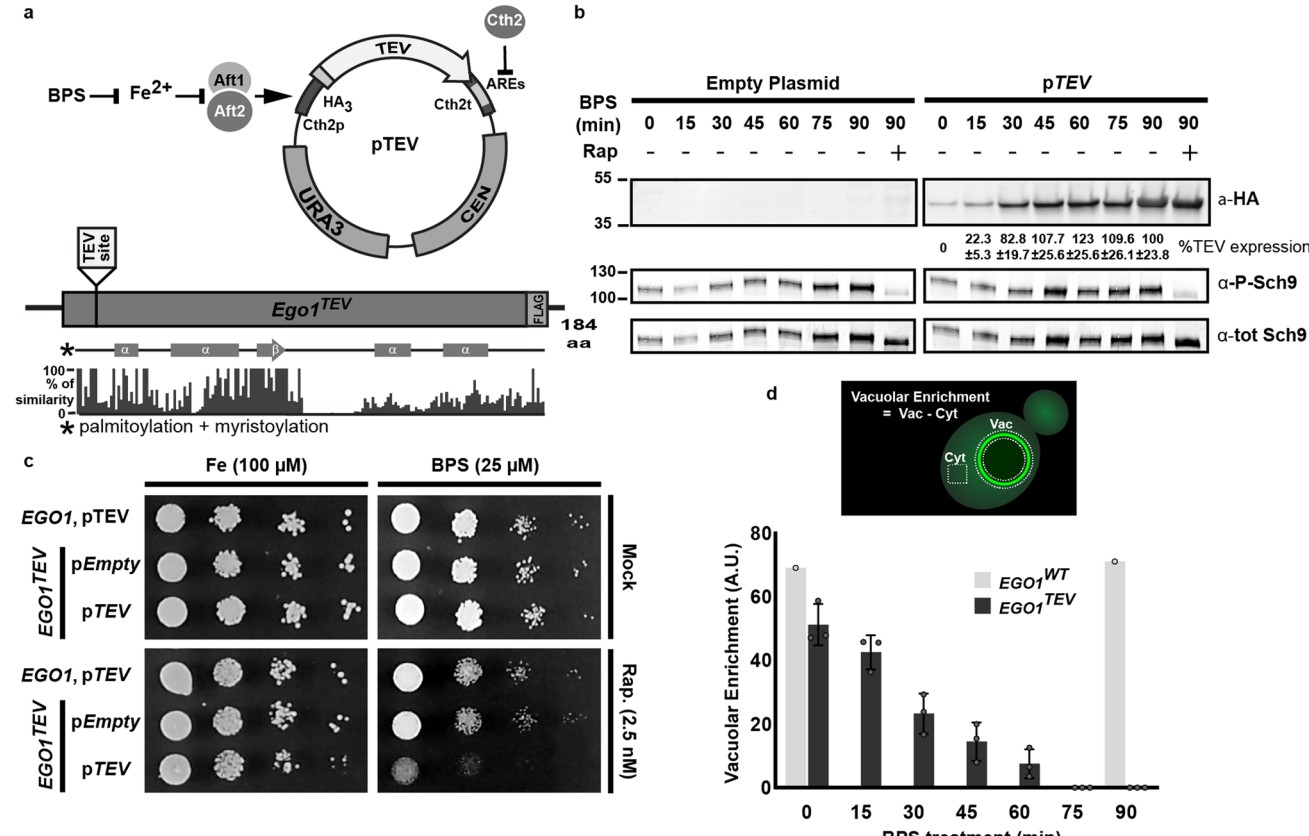

**Extended Data Fig. 1 | Inducible in vivo proteolysis of EGOC technical set up. a)** The HA tagged TEV protease coding sequence has been cloned under the control of the *CTH2* promoter and terminator. The terminator contains AU Rich Elements (AREs) recognized by the Cth2 protein that, when bound, destabilizes the transcript triggering its fast turnover. The promoter is additionally activated by Aft1 and Aft2 transcription factors which are repressed by the presence of Iron. Bathophenantroline dissulfonic acid is a specific Iron chelator used to quickly induce the TEV protease expression. The TEV site has been inserted into the N-terminal non-conserved sequence of Ego1 downstream of the lipidated residues. This expression system, unlike other, commonly used inducible promoters (such as the *GAL1* promoter), was chosen as it does not *per se* effect TORC1 signalling. **b)** Western Blot analyses of TEV expression (α-HA tag) and TORC1 activity were monitored by anti-Sch9 Ser758 phosphorylation (α-P-Sch9) and anti-total Sch9 (α-tot-Sch9) antibodies. *WT* cells transformed with empty or TEV plasmid (*pTEV*) were subjected to BPS and rapamycin treatments as indicated. Cells were harvested at the indicated time points. Averages of the TEV expression normalized to the maximum are indicated with the standard deviations for N = 3 replicates. The induction half-time is 23 minutes (EC50 Pearson's coefficient = 0.989). **c)** Serial 10-fold dilutions of strains of the indicated genotype were spotted onto plates containing vehicle or 2.5 nM rapamycin and 100 μM Mohr salt or 25 μM BPS. **d)** Bar chart representing the quantity of Ego3 present around the vacuolar membrane. EGOC depletion from the vacuolar membrane was determined by measuring the normalized Vacuolar Enrichment (see graphic): vacuole associated Ego3-GFP signal minus cytosolic background signal from Fig. 1b) Vacuolar depletion half-time is 25.5 minutes after TEV protease induction in a Ego1^TEV strain. The measures are based on n = 3 independent experiments with 10 cells analysed per replicate.

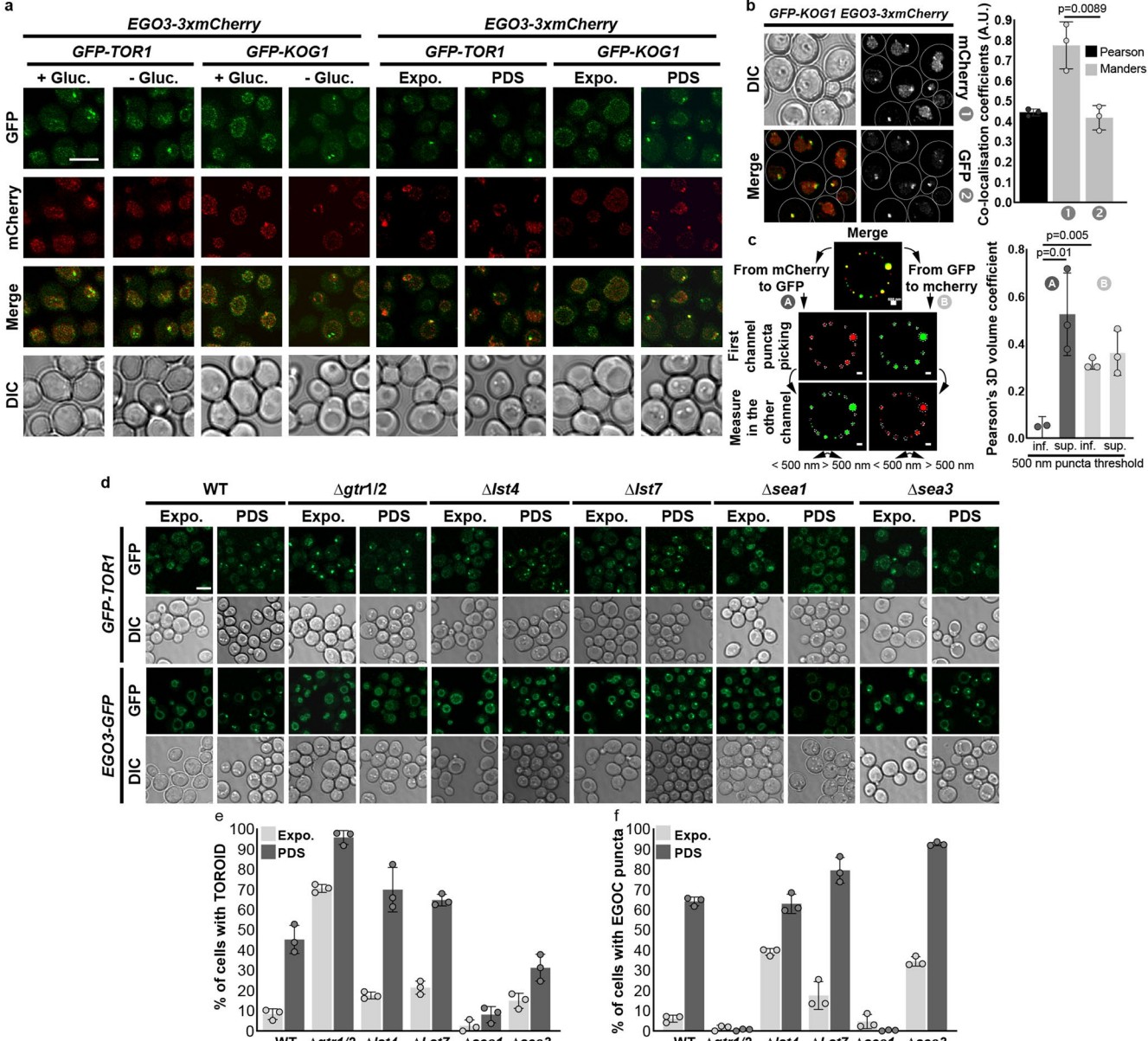

**Extended Data Fig. 2 | EGOC puncta formation requires TOROIDs and is regulated by Gtr-GAPs. a)** Cells expressing *GFP-KOG1* or *GFP-TOR1*, and *EGO3-3xmCherry* were imaged in exponential phase, after 30 minutes of acute glucose depletion or in PDS by confocal microscopy. (Scale bar, 5 µm) **b)** Confocal images of PDS GFP-KOG1 EGO3-3xmCherry-expressing cells were used to assess EGOC puncta and TOROID colocalization. Global 3D Pearson coefficient is indicated in black and reflects the global colocalization. Manders coefficients using mCherry (1) or GFP (2) channels are indicated in gray and reflects the overlaps of GFP in mCherry and mCherry in GFP signals, respectively. These parameters were calculated using Imaris software. The data represented on the bar charts were generated from n = 3 independent experiments. **c)** Workflow of the image analysis used to calculate the extent of TOROID colocalization with EGOC

puncta and *vice versa*, A and B respectively. Cells were segmented using an *in house* designed FIJI macro and Imaris 3D analyses were performed to segment puncta in each channel according to their size (inferior or superior to 500 nm). The bar chart shows the subsequent Pearson's coefficients from each channel over 3 independent experiments with at least 50 cells examined per replicate. **d)** WT, Δ*gtr1* Δ*gtr2, Δlst4, Δlst7, Δsea1 and Δsea3* strains expressing *GFP-TOR1* or *EGO3-GFP* were imaged in exponential phase or in PDS by confocal microscopy. (Scale bar, 5 µm) **e)** Bar chart representing the quantification of TOROID from **d)** with n = 3 independent experiments and at least 49 cells examined per replicate. **f)** Bar chart representing the quantification of EGOC dots from **d)** with n = 3 independent experiments and at least 59 cells examined per replicate.

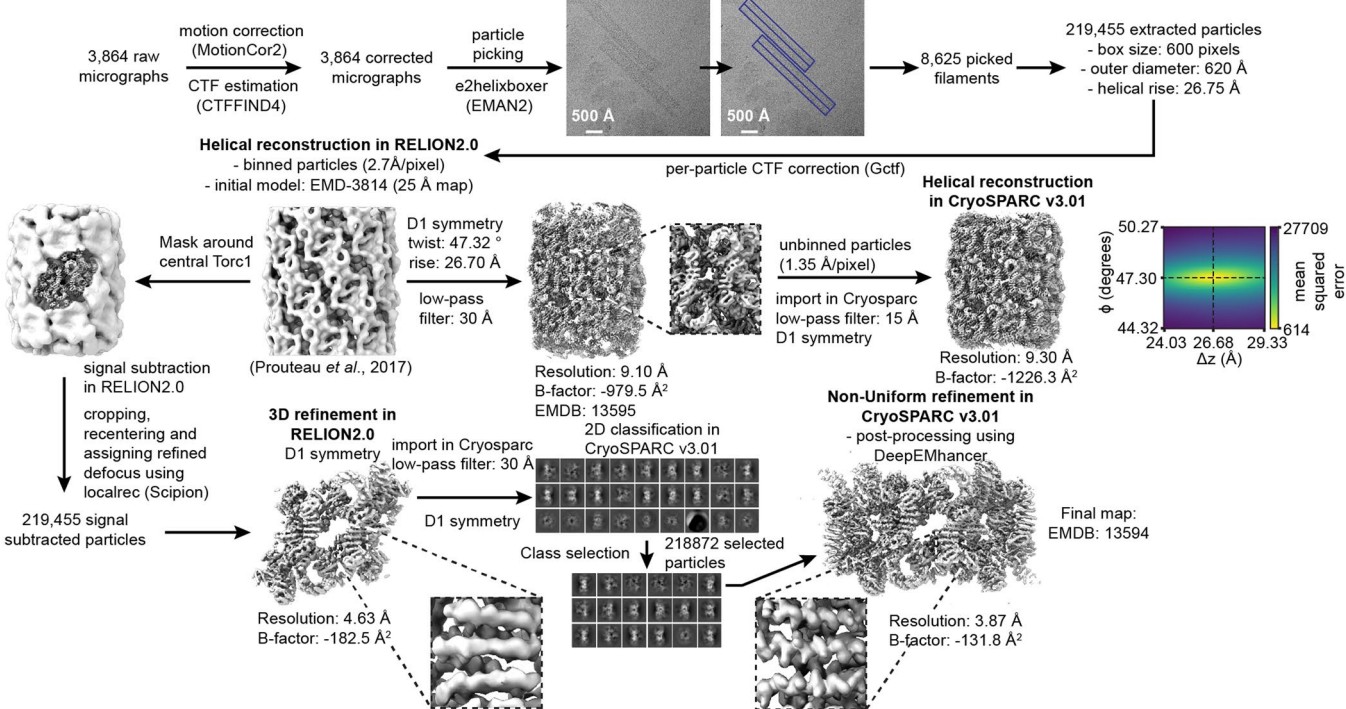

**Extended Data Fig. 3 | Processing workflow for TOROID structure determination by cryo-EM.** Processing workflow for TOROID structure determination by helical reconstruction and signal subtraction followed by single particle analysis (SPA). Software packages used during the different steps are indicated. The plot on the right shows the variation in estimated helical twist (in degrees) and rise (in Å) of individual TOROID segments during helical reconstruction. A summary of data collection/processing parameters, and refinement/validation statistics can be found in Table 1.

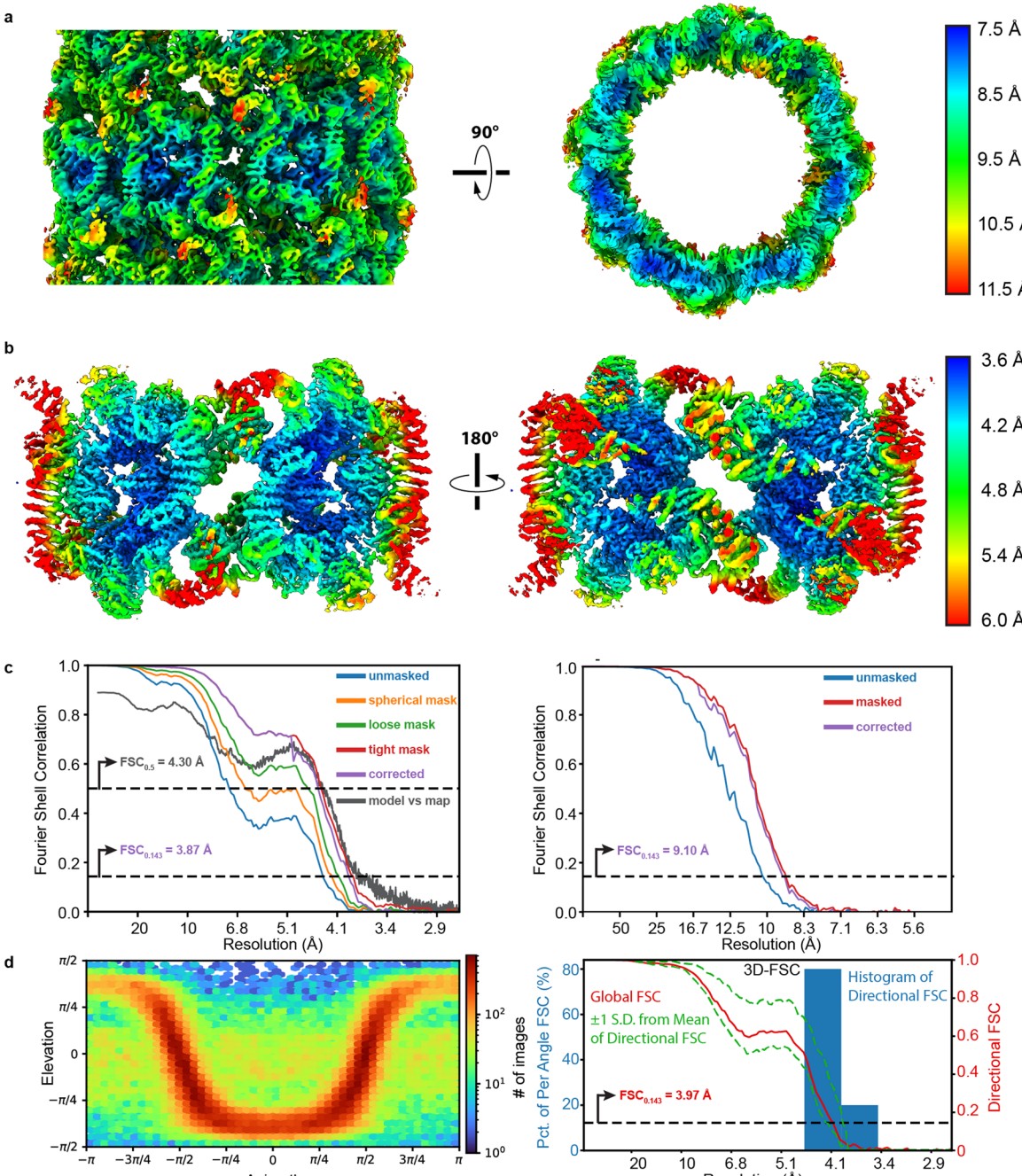

**Extended Data Fig. 4 | TOROID structure determination by helical reconstruction and signal subtraction of Cryo-EM data. a)** and **b)** Local resolution colouring of 3D reconstructions of TOROID by helical reconstruction (b) and signal subtraction and SPA (c). **c)** Gold-standard FSC curves for the TOROID map obtained after signal subtraction and SPA (left) or helical reconstruction (right). FSC curves are shown after applying either no mask (blue), a soft spherical mask (orange), a loose mask (green) or a tight mask (red) to both half maps prior to calculating the FSC. The corrected FSC curve (purple) is calculated by using the tight mask with correction by noise substitution. The estimated resolution at FSC = 0.143 is shown for the corrected FSC curves (purple). For the TOROID map obtained after signal subtraction and SPA, the model vs map FCS curve is shown in dark gray, as well as the resolution at FSC = 0.5. **d)** Elevation/Azimuth orientation distribution plot (left) and 3D-FSC plot (right) for the final 3D reconstruction of TOROID after signal subtraction and SPA.

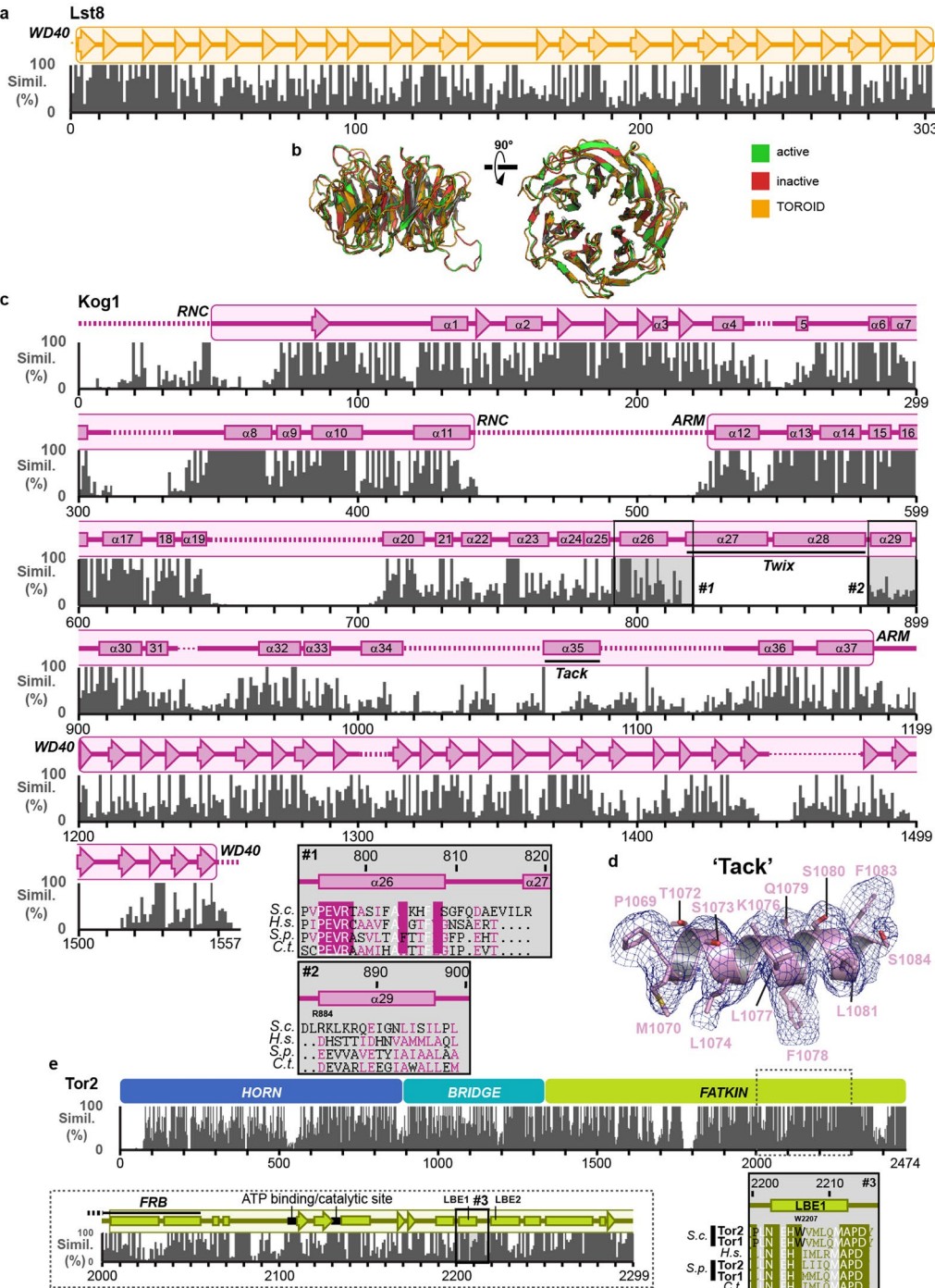

**Extended Data Fig. 5 | Evolutionary conservation of TORC1 components.**
**a)** Lst8 secondary structures from the TORC1 molecular model (Arrows: Beta-strands) aligned with the similarity sequence diagram based on the Lst8 orthologues aligned with ClustalX2. **b)** Side and top cartoon views of a structural alignment of Lst8 in TOROID (orange) with Lst8 from inactive mTORC1 (red, PDB ID: 6BCX) and Lst8 from active TORC1 bound to Rheb (green, PDB ID: 6BCU). Subunits are coloured according to the scheme shown below the panel. **c)** Kog1 secondary structures (Arrows: Beta-strands, Rectangles: alpha helices, dash lines: unstructured loops) and domains (RNC, ARM WD40) from the TORC1 molecular model aligned with the similarity sequence diagram based on the Kog1/Raptor

orthologues aligned with ClustalX2. Zoom on the α26 to α29 helix alignments show that Kog1 R884 is substituted by acidic residues in other organisms. **d)** Fit of the TOROID 'Tack' α-helix, proposed to correspond to residues 1,069-1,086 of Kog1, in the corresponding carved-out region of the TOROID Cryo-EM map. The helix is shown as a cartoon with labelled side chains shown as sticks. **e)** Tor2 domains (HORN, BRIDGE and FATKIN) aligned to the similarity sequence diagram based on the paralogues and orthologues aligned with ClustalX2. A closer look at the alignment within the FATKIN reveals that the W2207 of the first alpha helix of the Lst8 binding Element of Tor2 (LBE1) is substituted by basic residues in other organisms.

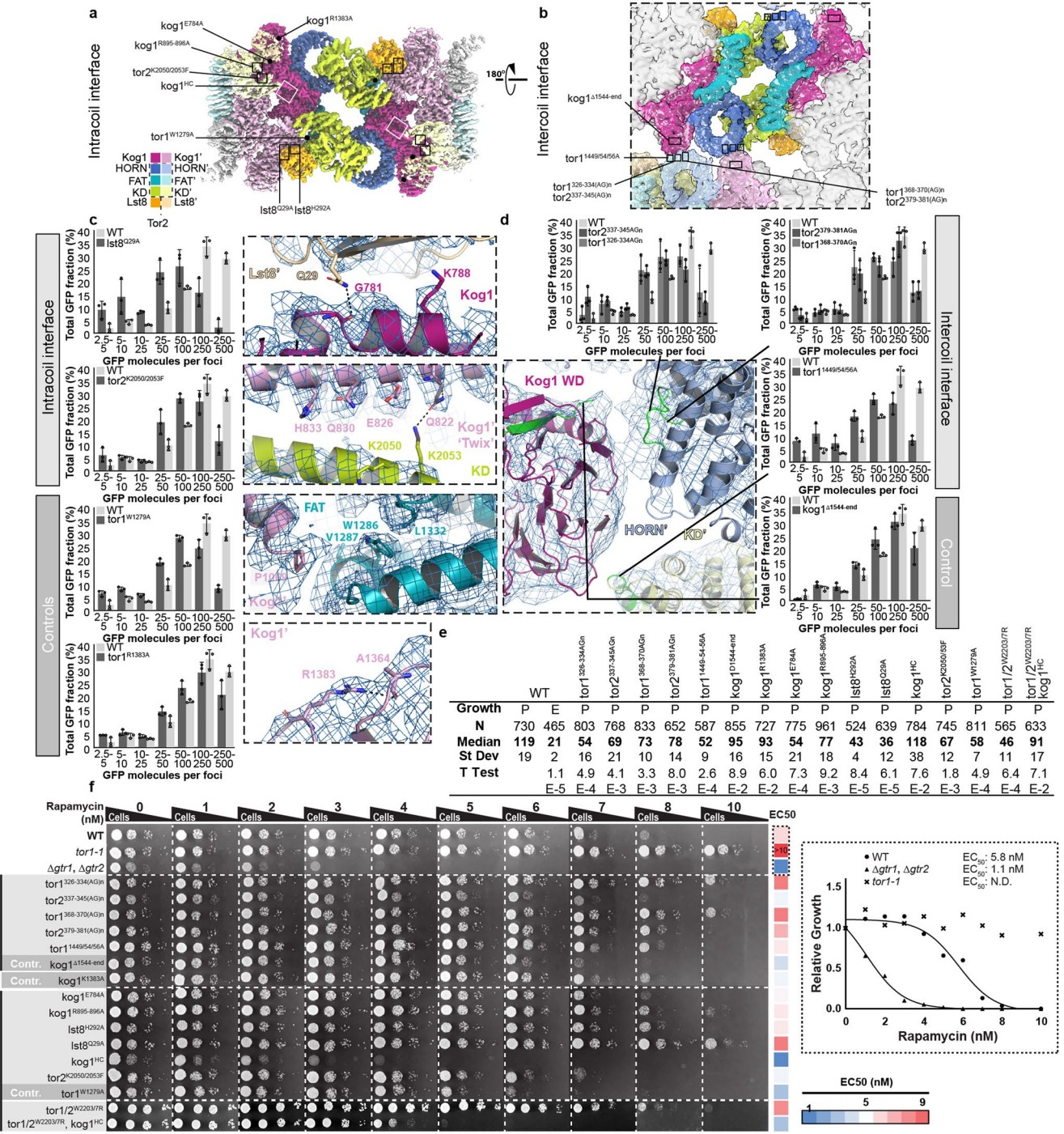

**Extended Data Fig. 6 | Additional mutations to test intra- and inter-coil TORC1-TORC1 interactions. a)** TORC1 molecular model extracted from the TOROID Cryo-EM map. The different TORC1 subunits are coloured as indicated. Adjacent protomers are shown in lighter shades. Locations of amino acid substitutions chosen to probe intra-coil interactions are indicated with black boxes. **b)** Same TORC1 molecular model as a), rotated 180 degrees. Locations of amino acid substitutions chosen to probe inter-coil interactions are indicated with black boxes. **c)** Left panel: Distribution of GFP molecules in TORC1 foci measured in PDS cells harbouring the indicated mutations. Graphs were generated based on the analysis of at least 315 cells per replicate over n = 3 independent experiments. The same control data are used in all graphs. Right insets: Zooms showing the different intra-coil interaction interfaces. **d)**

Same as c), but for $tor1^{326-333,334AGn}$ $tor2^{337-344,345AGn}$, $tor1^{368,369,370AGn}$ $tor2^{379,380,381AGn}$, $tor1^{1449/54/56A}$ and $kog1^{\Delta1544-end}$ mutants presumed to compromise inter-coil interactions. **e)** Statistical analyses of the TORC1 foci size distributions described in c) and d). GFP-Tor1 WT and mutated strains were imaged in exponential phase (E) or in PDS (P). Median and standard deviations were calculated over 3 independent experiments with at least 315 cells analyses per replicate. Two-sided Student-test were extracted between WT and mutants in PDS (p-values in table). **f)** Serial 10-fold dilutions of cells of the indicated genotypes were spotted onto plates containing the indicated concentrations of Rapamycin. The relative growth was measured and plotted for each strain; the corresponding $EC_{50}$ were extracted and quantified at the right of the panel. The $EC_{50}$ values vary from 1 (dark blue) to 9 nM (dark red).

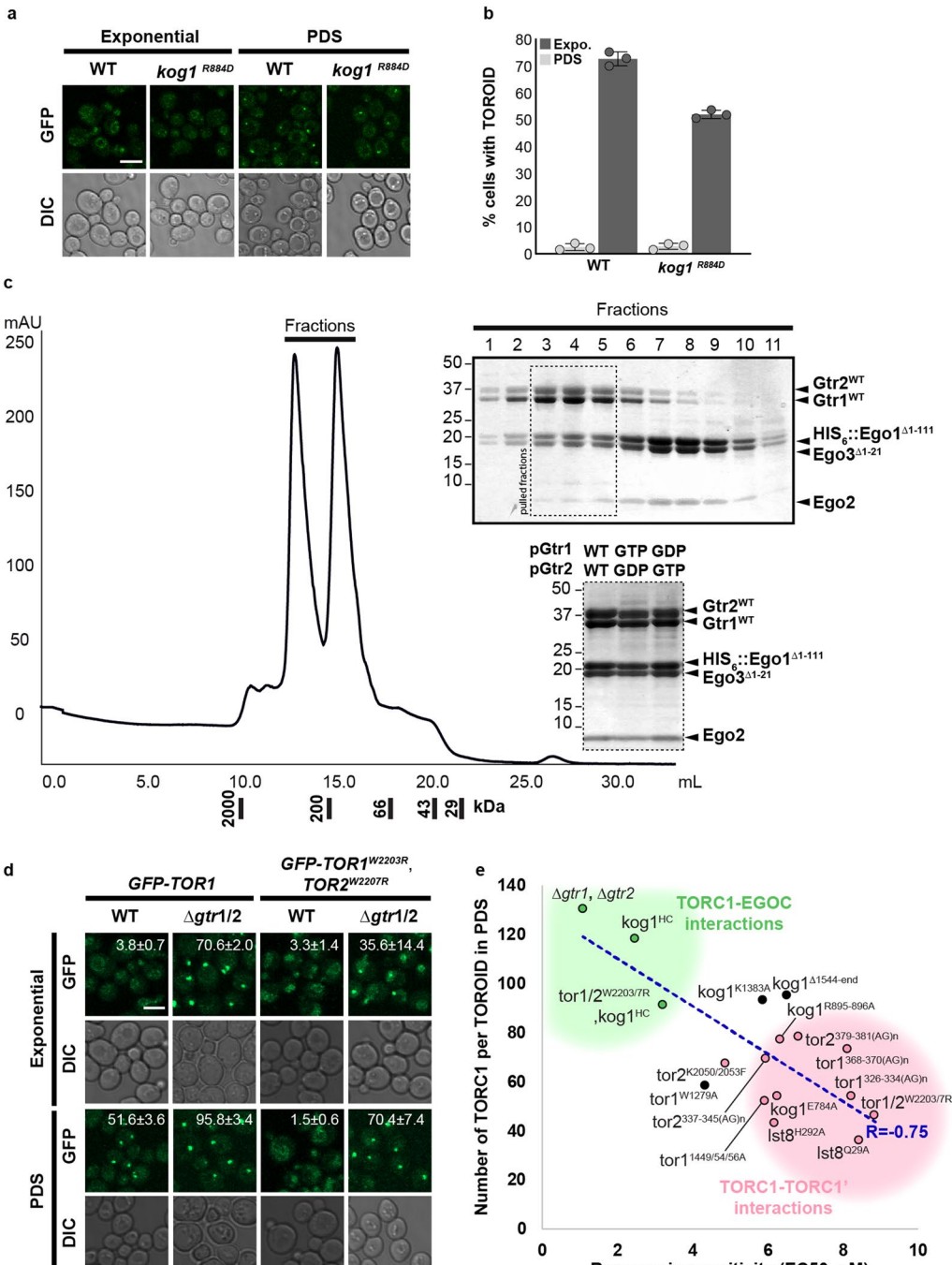

**Extended Data Fig. 7 | TOROID size is anti-correlated with rapamycin sensitivity. a)** *GFP-TOR1*, *GFP-TOR1 KOG1^{R884D}* cells were imaged in exponential phase or in PDS by confocal microscopy. (Scale bar, 5 µm) **b)** Bar chart representing the quantification of TOROID formation from **a)** with n = 3 independent experiments and at least 102 cells examined per replicate. **c)** Gel filtration profile and corresponding SDS–polyacrylamide gel electrophoresis of the five subunits of EGOC^{WT} (Ego1, Ego2, Ego3, Gtr1^{WT} and Gtr2^{WT}) as visualized with Coomassie Blue staining. Fractions 3-5 contain the fully assembled complex and were pooled together. The coomassie blue-stained polyacrylamide gel (bottom right) shows the different forms of EGOC: WT (Gtr1^{WT}–Gtr2^{WT}), 'active' (Gtr1^{GTP}–Gtr2^{GDP}) and 'inactive' (Gtr1^{GDP}–Gtr2^{GTP}). **d)** *GFP-TOR1*, *GFP-TOR1 gtr1 gtr2*, *GFP-tor1^{W2203R} tor2^{W2207R}* and *GFP-tor1^{W2203R} tor2^{W2207R} gtr1 gtr2* cells were imaged in exponential phase or in PDS by confocal microscopy. (Scale bar, 5 µm) **e)** Dot plot representing TOROID average size as a function of Rapamycin sensitivity of all the strains in S5F. The slope represents the linear regression with the Pearson coefficient (R).

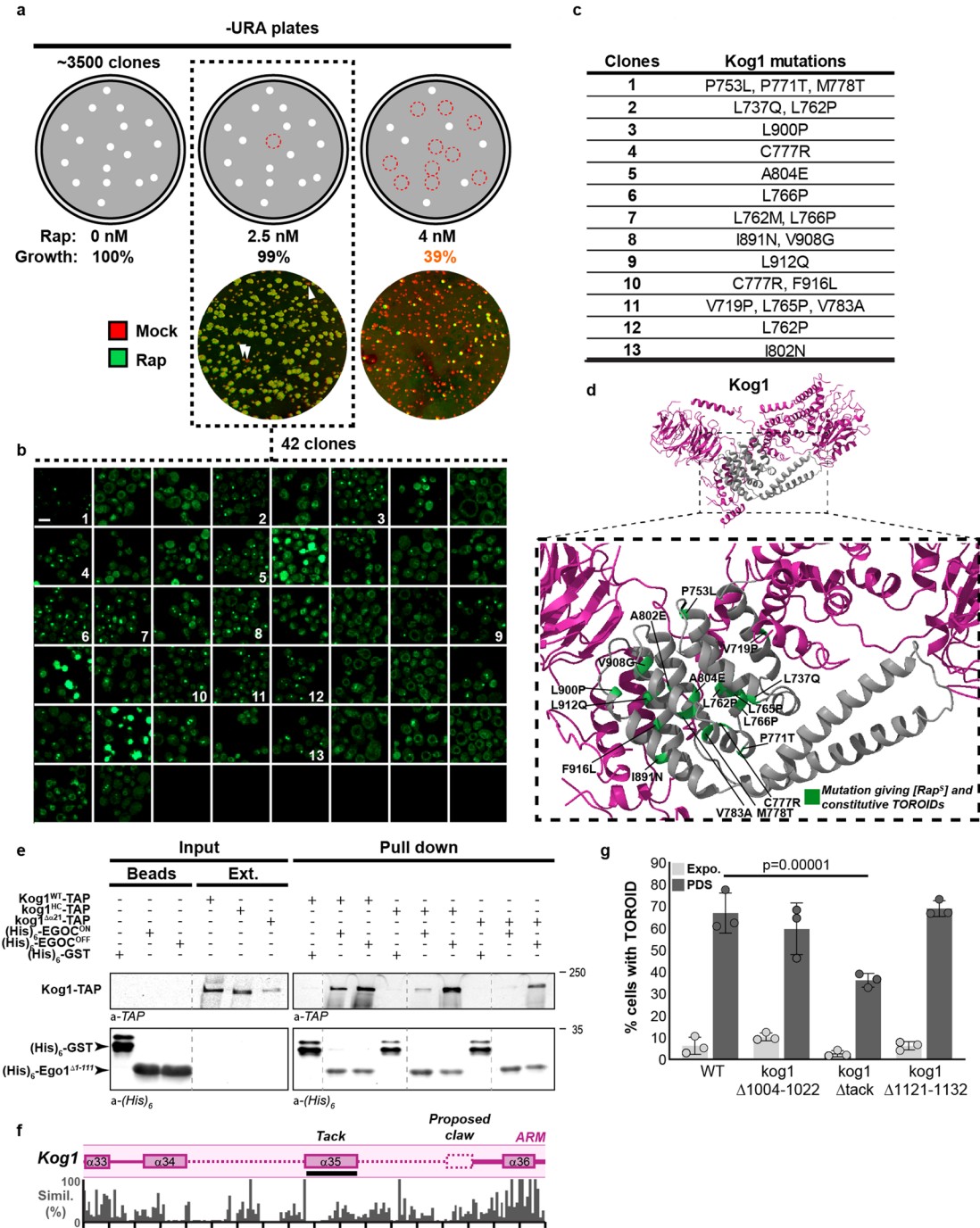

**Extended Data Fig. 8 | CrispR mutagenesis screen reveals mutations in the Armadillo domain of Kog1 involved in the TORC1-EGOC interaction.** Workflow of the CrispR mutagenesis screen. **a)** The screen is based on generation of mutagenized linear DNA fragments through repetitive rounds of error-prone PCR, introduced into *GFP-TOR1* yeast cells with CrispR-Cas9. ~3500 primary transformants were replicated from SC-URA plates to SC-URA containing 2.5 or 4 nM rapamycin. 99% of the clones grew in the presence of 2.5 nM Rapamycin, whereas only 39% grew in the presence of 4 nM rapamycin. **b)** Cells from 42 clones presenting hypersensitivity to 2.5 nM rapamycin were imaged in exponential phase by confocal microscopy. Clones presenting hypersensitivity to Rapamycin, constitutive TOROIDs and harboring punctual mutation(s) are labelled (from 1 to 12). (Scale bar, 5 μm) **c)** Table listing mutations identified in the labelled clones from b). **d)** Kog1 structural model (Upper panel). The region colored in grey (aa

717-895) was mutagenized by error-prone PCR. Lower panel: Zoom showing the locations of the altered amino acids that confer rapamycin-sensitivity and constitutive TOROIDs. **e)** *In vitro* pull downs of TORC1 containing either Kog1-TAP or kog1$^{HC}$-TAP or kog1$^{\Delta\alpha21}$-TAP using cobalt-Dynabeads pre-coated with purified (His)$_6$-proteins. Inputs and Pull Downs were analyzed by western blot. Gel splices are indicated with dotted lines. **f)** Zoom of the Kog1 secondary structure and alignment from S4C. Two additional features are added: a helix (α1004-1022) upstream the Tack and the putative claw (α1121-1132) **g)** Bar chart representing the percentage of *KOG1^{WT}*, *kog1^{Δα1004-1022}*, *kog1^{Δtack}* and *kog1^{Δα1121-1132}* cells displaying a TORC1 focus, marked with GFP-TOR1, in exponential phase or PDS. The analysis was performed based on n = 3 independent experiments with at least 47 cells examined per replicate.

# Reporting Summary

## Statistics

For all statistical analyses, confirm that the following items are present in the figure legend, table legend, main text, or Methods section.

| n/a | Confirmed | |
|---|---|---|
| ☐ | ☒ | The exact sample size ($n$) for each experimental group/condition, given as a discrete number and unit of measurement |
| ☐ | ☒ | A statement on whether measurements were taken from distinct samples or whether the same sample was measured repeatedly |
| ☐ | ☒ | The statistical test(s) used AND whether they are one- or two-sided<br>*Only common tests should be described solely by name; describe more complex techniques in the Methods section.* |
| ☐ | ☒ | A description of all covariates tested |
| ☒ | ☐ | A description of any assumptions or corrections, such as tests of normality and adjustment for multiple comparisons |
| ☐ | ☒ | A full description of the statistical parameters including central tendency (e.g. means) or other basic estimates (e.g. regression coefficient) AND variation (e.g. standard deviation) or associated estimates of uncertainty (e.g. confidence intervals) |
| ☐ | ☒ | For null hypothesis testing, the test statistic (e.g. $F$, $t$, $r$) with confidence intervals, effect sizes, degrees of freedom and $P$ value noted<br>*Give P values as exact values whenever suitable.* |
| ☒ | ☐ | For Bayesian analysis, information on the choice of priors and Markov chain Monte Carlo settings |
| ☒ | ☐ | For hierarchical and complex designs, identification of the appropriate level for tests and full reporting of outcomes |
| ☐ | ☒ | Estimates of effect sizes (e.g. Cohen's $d$, Pearson's $r$), indicating how they were calculated |

*Our web collection on statistics for biologists contains articles on many of the points above.*

## Software and code

Policy information about availability of computer code

| | |
|---|---|
| Data collection | Live-cell microscopy images were acquired either with LSM800 confocal microscope (Plan-Apochromat 63x/1.4 Oil Objective, Zeiss) or a TCS SP5 confocal microscope (Plan-Apochromat 63x/1.4 Oil Objective, Leica). Western blots were acquired with an Odyssey imager from a Li-Cor Odyssey 9120 imager. Cryo-EM images were collected on a Titan Krios microscope (Thermo Scientific), operated at 300 kV and equipped with a K2 summit direct electron detector (Gatan) camera operated in counting mode using SerialEM. |
| Data analysis | Western blot analysis: Software Image Studio version 5.2.5 from LI-COR; Light microscopy images processing: FIJI (ImageJ), Imaris version 9.8; Curve fit analysis: Prism 9.4.1 ; Cryo-EM: RELION 2.0, CTFFIND4, MotionCor2, EMAN2, Gctf, cryoSPARC 3.01; Structure analysis and vizualisation: USCF Chimera, USCF ChimeraX, PyMol 2.3; Model Building: DeepEMhancer, Alphafold. |

For manuscripts utilizing custom algorithms or software that are central to the research but not yet described in published literature, software must be made available to editors and reviewers. We strongly encourage code deposition in a community repository (e.g. GitHub). See the Nature Portfolio guidelines for submitting code & software for further information.

## Data

Policy information about availability of data

All manuscripts must include a data availability statement. This statement should provide the following information, where applicable:
- Accession codes, unique identifiers, or web links for publicly available datasets
- A description of any restrictions on data availability
- For clinical datasets or third party data, please ensure that the statement adheres to our policy

All light microscopy, western blots, spotassays and statistical analysis data are available at : 10.26037/yareta:kr6oskrjsneh5m2orna4zbqvo4.

## Human research participants

Policy information about studies involving human research participants and Sex and Gender in Research.

| | |
|---|---|
| Reporting on sex and gender | *Use the terms sex (biological attribute) and gender (shaped by social and cultural circumstances) carefully in order to avoid confusing both terms. Indicate if findings apply to only one sex or gender; describe whether sex and gender were considered in study design whether sex and/or gender was determined based on self-reporting or assigned and methods used. Provide in the source data disaggregated sex and gender data where this information has been collected, and consent has been obtained for sharing of individual-level data; provide overall numbers in this Reporting Summary. Please state if this information has not been collected. Report sex- and gender-based analyses where performed, justify reasons for lack of sex- and gender-based analysis.* |
| Population characteristics | *Describe the covariate-relevant population characteristics of the human research participants (e.g. age, genotypic information, past and current diagnosis and treatment categories). If you filled out the behavioural & social sciences study design questions and have nothing to add here, write "See above."* |
| Recruitment | *Describe how participants were recruited. Outline any potential self-selection bias or other biases that may be present and how these are likely to impact results.* |
| Ethics oversight | *Identify the organization(s) that approved the study protocol.* |

Note that full information on the approval of the study protocol must also be provided in the manuscript.

# Field-specific reporting

Please select the one below that is the best fit for your research. If you are not sure, read the appropriate sections before making your selection.

☒ Life sciences  ☐ Behavioural & social sciences  ☐ Ecological, evolutionary & environmental sciences

For a reference copy of the document with all sections, see nature.com/documents/nr-reporting-summary-flat.pdf

# Life sciences study design

All studies must disclose on these points even when the disclosure is negative.

| | |
|---|---|
| Sample size | No statistical methods were performed to predetermine the sample size, but the number of replicates for each experiment was based on our previous experience (Prouteau et al., 2017). Detailed information for the individual experiments including sample size and replicates are stated in the figure legends. For cryoEM studies, the number of particles used for each of the EM reconstructions has been stated in the methods section. |
| Data exclusions | For Cryo-EM, particles were sorted and selected as described in the "Methods" section. |
| Replication | At least triplicated experiments have been done for each data collection. all attempts at replication were successful. |
| Randomization | For cryoEM, extracted particles were randomly assigned to two separate groups to calculate half-maps and gold-standard FSC. |
| Blinding | All light microscopy experiments have been measured by blinded expimentators when it required manual counting. Automated measurements based on software were not blinded. For other experiments: no grouped samples. |

# Reporting for specific materials, systems and methods

We require information from authors about some types of materials, experimental systems and methods used in many studies. Here, indicate whether each material, system or method listed is relevant to your study. If you are not sure if a list item applies to your research, read the appropriate section before selecting a response.

## Materials & experimental systems

| n/a | Involved in the study |
|---|---|
| ☐ | ☒ Antibodies |
| ☒ | ☐ Eukaryotic cell lines |
| ☒ | ☐ Palaeontology and archaeology |
| ☒ | ☐ Animals and other organisms |
| ☒ | ☐ Clinical data |
| ☒ | ☐ Dual use research of concern |

## Methods

| n/a | Involved in the study |
|---|---|
| ☒ | ☐ ChIP-seq |
| ☒ | ☐ Flow cytometry |
| ☒ | ☐ MRI-based neuroimaging |

## Antibodies

Antibodies used — Mouse monoclonal anti-Sch9 p-S758 antibody, rabbit polyclonal anti-Sch9 antibody, mouse monoclonal anti-HA clone HA-7, ascites fluid (Sigma cat#H9658), mouse monoclonal anti-polyHistidine antibody, clone HIS-1 (Sigma cat#H1029-2ML), rabbit polyclonal anti-TAP Antibody (Open Biosystems cat#CAB1001).

Validation — Both Sch9 antibodies were made and already validated by our lab (Gaubitz et al., 2015; Prouteau et al., 2018). Anti-TAP and anti-HA antibodies were validated in papers from our lab (Binda et al., 2009; Bourgoint et al., 2018). Anti-polyhistidine antibody has been validated in numerous papers including Rodríguez-Escudero et al., 2018.

