## [Peer Review File · Nature Structural & Molecular Biology]

Peer Review Information

Manuscript Title: EGOc inhibits TOROID polymerization by structurally activating TORC1

Corresponding author name(s): Robbie Loewith, Manoël Prouteau, Jan Felix

Reviewer Comments & Decisions:

Decision Letter, initial version:

Message: 1st Mar 2022

Dear Dr. Loewith,

Thank you again for submitting your manuscript "EGOC inhibits TOROID polymerization by structurally activating TORC1". I apologize for the delay while we awaited the comments (copied below) from the 3 reviewers who evaluated your paper. In light of their reports, we remain interested in your study and would like to see your response to the comments of the referees, in the form of a revised manuscript.

You will see that all 3 reviewers are quite positive about the potential interest of the findings, and that each suggests text revisions to make the nature of the scientific advance more accessible to a wider readership. Reviewers 1 and 3 each specify points that should be clarified or corrected, and offer suggestions to improve data presentation or discussion. Reviewer 2 finds that some scientific issues remain to be resolved, and requests additional experimentation to further support the interactions of Kog1 proposed to regulate kinase activity. Editorially, we agree that these suggestions would strengthen the findings, and ask that they be incorporated in a revised manuscript.

Please be sure to address/respond to all concerns of the referees in full in a point-by-point response and highlight all changes in the revised manuscript text file. If you have comments that are intended for editors only, please include those in a separate cover letter.

When revising the manuscript, please bear in mind the following guidelines for our Article format:

- abstract should be maximum 150 words, no references;
- main text is typically between 3,000 and 4,000 words, and should be organized as introduction, results (with subheadings) and discussion.
- display items (figures and tables): typically between 6 and 8. Please note that the structural table should be in main article.
- supplementary items: Supplementary or Extended Data Figures should be a maximum of 10; other supplementary items are Suppl Table, Note, Video, Data Set.
- uncropped images of gels and blots should be presented in Supplementary Data Set.

We expect to see your revised manuscript within 6 weeks. If you cannot send it within this time, please contact us to discuss an extension; we would still consider your revision, provided that no similar work has been accepted for publication at NSMB or published elsewhere.

Reporting Summary:

When submitting the revised version of your manuscript, please pay close attention to our [href="https://www.nature.com/nature-research/editorial-policies/image-integrity">Digital Image Integrity Guidelines. and to the following points below:](https://www.nature.com/nature-research/editorial-policies/image-integrity)

Please note that all key data shown in the main figures as cropped gels or blots should be presented in uncropped form, with molecular weight markers. These data can be aggregated into a single supplementary figure item. While these data can be displayed in

a relatively informal style, they must refer back to the relevant figures. These data should be submitted with the final revision, as source data, prior to acceptance, but you may want to start putting it together at this point.

Data availability: this journal strongly supports public availability of data. All data used in accepted papers should be available via a public data repository, or alternatively, as Supplementary Information. If data can only be shared on request, please explain why in your Data Availability Statement, and also in the correspondence with your editor. Please note that for some data types, deposition in a public repository is mandatory - more information on our data deposition policies and available repositories can be found below: <https://www.nature.com/nature-research/editorial-policies/reporting-standards#availability-of-data>

We require deposition of coordinates (and, in the case of crystal structures, structure factors) into the Protein Data Bank with the designation of immediate release upon publication (HPUB). Electron microscopy-derived density maps and coordinate data must be deposited in EMDB and released upon publication. To avoid delays in publication, dataset accession numbers must be supplied with the final accepted manuscript and appropriate release dates must be indicated at the galley proof stage.

[Redacted]

We look forward to seeing the revised manuscript and thank you for the opportunity to

review your work.

With kind regards,

Beth

Beth Moorefield, Ph.D.
Senior Editor
Nature Structural & Molecular Biology

Referee expertise:

Referee #1: structural biology/TOR signaling

Referee #2: PI3K/TOR signaling

Referee #3: structural biology/PIK3 signaling

Reviewers' Comments:

Reviewer #1:

Remarks to the Author:

The TORC1 kinase complex is a central hub for regulating homeostasis and growth in budding yeast in response to nutrient availability. Its human equivalent is mTORC1, which additionally responds to growth factor signals and differs from TORC1 in several aspects, e.g. in input sensing and localization. In this manuscript the team of R. Loewith and collaborators provide a detailed structural analysis of an inhibited filamentous form of TORC1, referred to as TOROID, and the role of the EGO complex (EGOC, related to the human Ragulator-RagA/B-RagC/D complex) consisting of EGO1-3 (EGO-TC) and the Gtr1&Gtr2 GTPases in controlling TOROID formation. This work follows up on earlier work of the same team that reported the discovery of TOROID filaments, a low-resolution structural analysis of this filaments as well as a connection via an unknown mechanism between Gtr1 and Gtr2 and TOROID assembly.

In the current work, the authors show that induced engineered loss of EGOC from the vacuole triggers TOROID formation, while re-establishing EGOC at the vacuole causes TOROID dissolution. They observe interactions of both, active and inactive nucleotide states of Gtr1/Gtr2 with TORC1, with the inactive state (Gtr1-GDP/Gtr2-GTP) favoring TOROID formation and the active state (Gtr1-GTP/Gtr2-GDP) disfavoring it, and a partial correlation of EGOC and TOROID punctae formation.

To improve the resolution of TOROID analysis by cryo-EM, the authors now turn to a single-particle analysis mode that provides a 3.8Å resolution reconstruction of a TOROID fragment, which - together with available structural data on (m)TORC1 and Rag-type GTPases - allows a detailed discussion of TOROID structure. Application of this approach is fully appropriate for this analysis and is further justified by the finding that TOROIDS consist of flexibly linked stacked coils. Structural data are of high quality, and despite the limited resolution well supporting the analysis presented here. In TOROIDS, TORC1 is in a highly open conformation, closer to non-activated open than to closed Rheb-activated mTORC1.

The authors further probe and discuss the relevance of regions in Kog1 for TOROID

formation and EGOC interaction and based on their structural data, mutational analysis and a random mutagenesis screen propose that EGOC binds to TORC1 similarly as Rag-Ragulator to mTORC1, albeit with differences in the detailed mode of interaction, and that the EGOC binding site overlaps with critical intermolecular contacts in TOROIDS. This would lead to a competition between EGOC binding and TOROID formation and provide a mechanism by which EGOC binding promotes TORC1 downstream signaling. Eventually, the results and discussion section could be focused more on most relevant findings to increase the accessibility for a general readership. The last parts of the discussion, albeit interesting, appear as speculative, considering the differences between human and yeast TORC1 interactions also found in this manuscript, and, without any novel data on human mTOR presented here, may fit better into a review-style publication. Overall, the findings presented in this manuscript are highly relevant for a mechanistic understanding of TORC1 regulation in budding yeast, and for further dissecting the differences between yeast and human (m)TOR regulation.

Comments:

The authors refer to D1 symmetry used in the reconstruction. Wouldn't this be more commonly referred to as C2 symmetry?

For RMSD values, I suggest reducing the number of significant digits.

In Fig. 3b, I suggest removing the dashed line, the symmetry symbol seems sufficient. In Fig. 3c, showing all side chains in the field of view may provide a more unbiased representation of map quality.

Fig. S3 lists 3864 raw micrographs at the start of processing, but the text lists collection of 4901 movies. Presumably the approx. 1000 missing movies were excluded based on early processing steps?

Fig. S3 lists 218872 particles selected for non-uniform refinement, but Supp. Table 1 lists 213808 particles finally used.

For real-space refinement into a D1 symmetric map, have the authors possibly used NCS constraints instead of NCS restraints?

Although the helical nature of filaments should provide a full set of views, please add a 3D-FSC and viewing direction plot to Fig.S3 for the single-particle reconstruction.

In the caption to Figure S3, is (b) and (c) mixed up? (b) should be "helical reconstruction" and (c) SPA.

In the local resolution map for single particle reconstruction, the drop in local resolution at the lateral ends is extreme (and may even look less so based on the box sizes used to calculate local resolution), despite an apparently large interaction interface in this direction in the TOROID filament. Is this only due to conformational variability or could the wall of the filament have holes or missing protomers? Did the authors try to use 3D-classification or variability analysis to identify subsets of the approx. 200'000 particles that are more homogenous in this region?

Page 9 top: "constitutively from TOROIDS" -> "constitutively form TOROIDS"

Page 11: add closing parentheses after Pib2.

Reviewer #2:

Remarks to the Author:

In this study Loewith and colleagues report on a follow-up from their earlier work, in which they had reported the formation of large assemblies of TORC1 referred to as TOROIDs in glucose-deprived yeast cells. Here the authors provide a greatly improved cryoEM structure of TOROID (at about 4Å resolution) that is combined with functional yeast genetic and imaging data to propose a novel model for the regulation of TOROID assembly and dissolution via the EGO complex. They use elegant yeast genetics to demonstrate that acute removal of EGO from the yeast vacuole induces TOROID formation and this correlates with impaired TORC1 activity evidenced by hypersensitivity to Rapamycin. Live imaging of endogenously tagged fluorescent Kog1/ Raptor and Ego3 suggests that EGO partially colocalizes with TOROID but EGO condensates are not required for TOROID assembly. Instead, it is confirmed that TOROIDs are largely controlled by Gtr/ Rag activity status; i.e. active Gtr/EGO counteracts TOROID formation, while inactive Gtrs promote TOROID assembly. The authors further present a high resolution 3D reconstruction of TOROID, in which the dimeric TORC1 building blocks appear to adopt an inactive conformation reminiscent of inactive mammalian mTORC1. Within the TOROID assembly intra-coil interactions between two adjacent TORC1 dimers can be traced to extensive interactions between Kog1/ Raptor in one and Tor2 in the other TORC1 unit. The main interface capitalizes on cation- π interactions via conserved Trp residues in Tor2 and a basic patch in Kog1. Surprisingly, the only unique elements (i.e. by comparison to mammalian Raptor) identified in Kog1 called the Twix region and the Tack helix are dispensable for TOROID assembly or binding to EGO. Instead, they may contribute to occlusion of the active site in Tor when assembled into TOROID. Finally, Raptor/ Kog1 swap mutants are combined with random mutagenesis to define an interface comprising helices 21,24,26, and 29 of yeast Kog1 that are involved in association with the EGO complex. Based on these results the authors propose that EGO modulates TOROID dynamics and thereby TORC1 activity in cells via Kog1 binding.

The data contained in this manuscript are of high quality and of interest to the community. Some of the findings are poorly supported by data as figures 3D, 7G and S7 (as quoted in the text) appear to be missing from the submission and thus render a proper assessment of this study difficult. I also found the paper hard to follow at times and noted an unusually large number of typographical errors, open brackets etc. These latter shortcomings can be corrected and improved in a suitably revised version of this paper. In addition, a number of important scientific questions remain to be resolved before publication of this work.

Major issues:

1. A key structural element of the study is the presence of unique intra-coil interactions between Kog1 and all other subunits of the adjacent monomer within the dimeric unit including the kinase domain itself. However, from the data presented it is unclear to me to which degree the identified interface truly renders Tor kinase inactive. A more thorough analysis of these interactions and their relationship to kinase activity is required, ideally by kinetic biochemical experiments involving mutant proteins. In this context: What is the role of the Tack helix interactions displayed in Fig 5a,ii on Tor kinase activity? As it stands, I found no real evidence for the functional importance of either the Twix or the Tack helices in the control of TORC1 function. Does TORC1 become hyperactive when either Twix or Tack are deleted/ mutated?

2. Both active and inactive EGOC are shown to associate with TORC1 and/or TOROID to different degrees. The authors presume that the Kog1 interface identified through mutagenesis associates with Gtr/ Rags but no direct evidence is provided. Furthermore, the authors speculate in the discussion that EGOC may bind to TORC1 via multiple modes. Is the same Kog1 interface used to associate with both active and inactive EGOC? What is the explanation for the enhanced binding of inactive Gtr/EGOC to TORC1 and TOROID compared to the weak association of active EGOC with TORC1?
3. The believed to be critical pi-cation interaction between Tor and Kog1 is only addressed by mutation of the two Trp in Tor while mutation of a single Arg in Kog1 appears to be without effect. Further biochemical evidence for the charge complementation model, e.g. by analysis of mutants of the entire basic patch would greatly strengthen the model and increase the impact of the study.
4. The distribution of figure panels between the main figures and the supplement could be improved. For example, Fig. S4C appears to be quite central and should be displayed as part of a main figure.

Reviewer #3:

Remarks to the Author:

The manuscript is an important contribution to understanding regulation of TOR. Prouteau et al. have described in unprecedented detail the structure of a large assembly of budding yeast TORC1 units that has been described as the TOROID. A previous report of this structure at low resolution was sufficient to gain a general impression of the arrangements of the units within the filaments that make up the TOROID. However, in the current manuscript they have improved the cryo-EM structure by doing a focused refinement with subtraction for each TORC1 subunit with its contacting units within the TOROID. Aligning all of these units gives a 3.8 Å resolution image of the intra-coil interactions. They conclude that the contacts within the TOROID block the active site of the TORC1 through interactions with the FRB, similarly to how FKBP/Rapamycin interacting with the FRB inactivates the enzyme. Their cellular observations are consistent with this being an inactive state of the TORC1, since light microscopy suggests formation of TOROID-containing puncta when glucose is restricted and TORC1 activity is downregulated.

In addition to the structural observations, the authors have also shown that the active EG1/2/3/Gtr1/2 (EGOC) complex interacts with TOROIDs, and this is consistent with the general picture that Loewith and his colleagues developed prior to this study: active Gtrs bind TORC1 and disassemble TOROIDs, because the surface on Kog1 that interacts with active Gtrs is a region that encompasses interactions between Kog1 and the TOR from an adjacent subunit in the TOROID. The current higher-resolution work fills in the molecular details of these interactions and verifies the structure by mutagenesis. However, it also shows that the picture is incomplete, because they convincingly show that the inactive EGOC also robustly binds TORC1. They propose that TORC1 is capable of making two different types of interactions with EGOC, depending on the nucleotide state of the Gtrs. Although mTORC1 assemblies analogous to TOROIDs have not been reported, it may be that similar assemblies occur in circumstances that have not yet been characterized. Such assemblies could give an additional layer of control to mTORC1. The manuscript will provide a useful framework for designing experiments to examine the possibility that mTORC1 could also form TOROID-like assemblies.

The yeast work is elegant and convincingly executed to illustrate the regulation of TOROIDS by EGOC and the glucose state. The structural work is innovative and informative.

There are some minor points that should be addressed in revision. These are summarized in the following.

lines 133-134

The authors start the results with "We previously proposed that TOROID regulation downstream of glucose-derived signals was mediated via the EGOC (27)"

The word "regulation" is sloppy and should be replaced. Is it up-regulation or down-regulation? Does EGOC enhance TOROIDS or repress TOROIDS? Reference 27 does make it clear what they proposed EGOC is doing (though it too is plagued with similar ambiguous writing). The sentence should be re-written to make it clear:

We proposed that EGOC interaction prevents formation of TORROIDS.

Lines 90-93

The authors state "TOROID formation was found to be dependent upon the two yeast RAG orthologues GTR1 (RAGA/B) and GTR2 (RAGC/D): delta gtr1 /delta gtr2 cells present constitutive TOROIDS even in the presence of glucose and are impaired in their ability to downregulate Sch9 phosphorylation upon glucose starvation."

The statement might be clear to geneticists or yeast biologists, but, given the journal, it should be rephrased to be a bit more accessible. The statement "TOROID formation was found to be dependent upon the two yeast RAG orthologues GTR1 (RAGA/B) and GTR2 (RAGC/D)..." sounds like the Gtrs promote assembly of TOROIDS, but active Gtrs, in fact, seem to prevent the assembly. The statement is technically correct. If Gtr1/2 prevents assembly, assembly is dependent on whether there is Gtr1/2 or not. However, it would be a little easier to follow if the authors state something like:

"TOROID formation was found to be inhibited by the two yeast RAG orthologues GTR1 (RAGA/B) and GTR2 (RAGC/D), because delta gtr1 /delta gtr2 cells form constitutive TOROIDS, even in the presence of glucose and do not downregulate Sch9 phosphorylation upon glucose starvation."

Lines 107-110:

The sentence "Indeed, expressing mutants that lock the Gtrs in an inactive, Gtr1GDP/Gtr2GTP conformation, TORC1 is concentrated in a TOROID punctum on the vacuolar membrane whereas in cells expressing active, Gtr1GTP/Gtr2GDP, TORC1 is diffusely localized over the surface of the vacuole 27."

Would read better as:

"Indeed, by expressing mutants that lock the Gtrs in an inactive, Gtr1GDP/Gtr2GTP conformation, TORC1 is concentrated in a TOROID punctum on the vacuolar membrane whereas by expressing active, Gtr1GTP/Gtr2GDP, TORC1 is diffusely localized over the surface of the vacuole 27."

(insert the word by)

Line 121

The authors state "We found that a central hub of interactions locks the Kog1 Armadillo (ARM) domain on top of the kinase domain of Tor2'and sterically occludes the kinase active site."

This term Tor2' has not yet been introduced in the text.

I would suggest "We found that a central hub of interactions locks the Kog1 Armadillo (ARM) domain on top of the kinase domain of the next Tor2 molecule in the TOROID (Tor2') to sterically occlude the kinase active site."

Lines 124-125

The authors say "this same hub is engaged by the EGOC to mediate glucose-dependent regulation of TORC1 signalling."

I understand that regulation is a neutral term. It could mean activate or inactivate. The text would be much more dynamic and readable, if the authors take a position and say what they mean. I would suggest:

"this same hub is engaged by EGOC to activate TORC1 signalling in response to glucose."

Lines 125-126

"we propose that the EGOC is necessary and sufficient for TOROID regulation through competition for a common binding hub in Kog1"

Avoiding the neutral, boring and less informative word "regulation", it would be better to say something like:

we propose that the EGOC is necessary and sufficient to prevent TOROID assembly, by competing for a common binding hub in Kog1.

lines 127-129:

"trans binding of this hub to Tor1'/2' sequesters TORC1 into an inactive TOROID, while binding to active EGOC liberates TORC1 dimers able to signal to downstream targets."

Again, the authors are using a term Tor1'/Tor2' that has not yet been introduced. What does "trans" binding mean? This has not been defined in the text at this point.

In Fig 3, the domain that the authors refer to as the FAT appears to be what has been referred to as the M-HEAT or bridge in previous papers. The region referred to as the kinase domain appears to be the FAT and kinase domain (also known as the FATKIN). The authors should the FAT and kinase domain consistent with previous publications.

In Fig. 4 and in text lines 283 and 284. mTORC1 without RHEB is referred to as "inactive". This is misleading. The enzyme activity of this complex without has been characterized by several groups. It is more appropriate to refer to this as basal state or apo-state.

The authors should state clearly how they measure the angles illustrated in Figs 4A,B. There appears to be nothing in the methods.

In Fig 3 and lines 295,296, the authors refer to Tor2'. Is this Tor2'in an adjacent unit in the Toroid, or is it the other Tor2 in the Torc2 dimer?

It would be unambiguous if the authors change "...interacts with the FRB' domain of Tor2' in an adjacent TORC1' subunit. " to

It would be unambiguous if the authors change "...interacts with the FRB' domain of Tor2' in an adjacent TORC1' in the TOROID. "

The confusion arises because TORC1 is not a subunit. It is a complex.

Lines 311-313

"The Twix and the Tack shield the kinase active site and thereby block substrate access, providing, together with the large allosteric change described above, a structural explanation for why TORC1 in TOROIDS is inactive (Figure 4D, Figure S4C).|"

First. There should be a clear picture of where they would expect a substrate peptide

interacting with the FRB/Kog1 interface in the TOROID. There is a region circled called "active site", but no explanation as to what this is or how it is deduced. In Fig S4C, there is a region colored green in the active site, but the legend does not explain what this is. Is this the ATP-binding site or the peptide-binding site on the FRB?
 Second. There has been no allosteric change described above. The only allosteric change described above is the conformational change between apo and RHEB-bound mTORC1 conformations. There is a statement that the TOROID is more open than the mTORC1 dimer, but this is not an allosteric change. The proper comparison to infer allostery would be to the TORC1 dimer.

Lines 463-465

"Replacement of the proximal alpha-helix 34 (Kog1 1004-1022) had no effect on TOROID regulation, while replacement of the entire Tack helix (Kog1 1121-1131) leads to a mild, but significant, impairment of TOROID formation (Figure S7E, F)."
 There appears to be a typo here. The Tack helix is 1069-1086. Not 1121-1131 (this is the claw).

Lines 497-498

"a dynamic equilibrium between active, free TORC2 dimers 497 and inactive TOROID polymers in a cell"

This is probably a typo. TORC2 should be TORC1. This is the only place in the manuscript that TORC2 has been mentioned.

Author Rebuttal to Initial comments

Dear Beth,

We are pleased to return our revised manuscript NSMB-A45852. The text has been edited extensively to meet the word limit (presently 4820 words intro/results/discussion).

Below are our point-by-point replies to the reviewers' comments. I think we have managed to address all concerns. We hope that these revisions will satisfy you and the three helpful reviewers.

Sincerely,
 Robbie

Reviewers' Comments:

Reviewer #1:

Remarks to the Author:

The TORC1 kinase complex is a central hub for regulating homeostasis and growth in budding yeast in response to nutrient availability. Its human equivalent is mTORC1, which

additionally responses to growth factor signals and differs from TORC1 in several aspects, e.g. in input sensing and localization. In this manuscript the team of R. Loewith and collaborators provide a detailed structural analysis of an inhibited filamentous form of TORC1, referred to as TOROID, and the role of the EGO complex (EGOC, related to the human Regulator-RagA/B-RagC/D complex) consisting of EGO1-3 (EGO-TC) and the Gtr1&Gtr2 GTPases in controlling TOROID formation. This work follows up on earlier work of the same team that reported the discovery of TOROID filaments, a low-resolution structural analysis of this filaments as well as a connection via an unknown mechanism between Gtr1 and Gtr2 and TOROID assembly. In the current work, the authors show that induced engineered loss of EGOC from the vacuole triggers TOROID formation, while re-establishing EGOC at the vacuole causes TOROID dissolution. They observe interactions of both, active and inactive nucleotide states of Gtr1/Gtr2 with TORC1, with the inactive state (Gtr1-GDP/Gtr2-GTP) favoring TOROID formation and the active state (Gtr1-GTP/Gtr2-GDP) disfavoring it, and a partial correlation of EGOC and TOROID punctae formation. To improve the resolution of TOROID analysis by cryo-EM, the authors now turn to a single-particle analysis mode that provides a 3.8Å resolution reconstruction of a TOROID fragment, which - together with available structural data on (m)TORC1 and Rag-type GTPases – allows a detailed discussion of TOROID structure. Application of this approach is fully appropriate for this analysis and is further justified by the finding that TOROIDS consist of flexibly linked stacked coils. Structural data are of high quality, and despite the limited resolution well supporting the analysis presented here. In TOROIDS, TORC1 is in a highly open conformation, closer to non-activated open than to closed Rheb-activated mTORC1.

The authors further probe and discuss the relevance of regions in Kog1 for TOROID formation and EGOC interaction and based on their structural data, mutational analysis and a random mutagenesis screen propose that EGOC binds to TORC1 similarly as Rag-Regulator to mTORC1, albeit with differences in the detailed mode of interaction, and that the EGOC binding site overlaps with critical intermolecular contacts in TOROIDS. This would lead to a competition between EGOC binding and TOROID formation and provide a mechanism by which EGOC binding promotes TORC1 downstream signaling.

Eventually, the results and discussion section could be focused more on most relevant findings to increase the accessibility for a general readership. The last parts of the discussion, albeit interesting, appear as speculative, considering the differences between human and yeast TORC1 interactions also found in this manuscript, and, without any novel data on human mTOR presented here, may fit better into a review-style publication.

Overall, the findings presented in this manuscript are highly relevant for a mechanistic understanding of TORC1 regulation in budding yeast, and for further dissecting the differences between yeast and human (m)TOR regulation.

Response to reviewer:

We thank the reviewer for his/her constructive feedback. Our revised manuscript has been much shortened, including removal of the admittedly too speculate aspects of the discussion. Specific responses to the reviewer's comments follow below.

Comments:

The authors refer to D1 symmetry used in the reconstruction. Wouldn't this be more commonly referred to as C2 symmetry?

D1 symmetry in helical reconstruction corresponds to a single 2-fold axis perpendicular to the helical (z-)axis, and no rotational symmetry along the helical axis, thus contrasting with C_n helical symmetry where an additional n -fold rotational axis along z is present (He S. and Scheres S.H.W., 2017, Helical reconstruction in RELION, Journal of Structural Biology, 198:3, 163-176). After signal subtraction, we did not alter the orientation of the extracted particles with respect to the z -axis ('Do symmetry alignment' option in CryoSPARC), and thus kept D1 symmetry during 3D refinement. Indeed, for 3D refinement of the signal subtracted particles, D1 symmetry and C2 symmetry are essentially identical, except for the orientation of the 2-fold axis, which is perpendicular to the z -axis for D1 symmetry, and along the z -axis for C2 symmetry.

For RMSD values, I suggest reducing the number of significant digits. In Fig. 3b, I suggest removing the dashed line, the symmetry symbol seems sufficient. In Fig. 3c, showing all side chains in the field of view may provide a more unbiased representation of map quality.

RMSD values are changed in the main text to 1 digit after the decimal point.

The requested changes to the figure panels have been done.

Fig. S3 lists 3864 raw micrographs at the start of processing, but the text lists collection of 4901 movies. Presumably the approx. 1000 missing movies were excluded based on early processing steps?

Data was collected from Lacey grids with irregular holes, and the presence of TOROID filaments was dependent on the location on the grid. Therefore, not all movies contained TOROID filaments, and 'empty' movies were discarded from further processing. We added following to the Methods section:

'TORC1 filaments were picked manually using the e2heliboxer module in EMAN2⁵⁹ from a total of 3864 micrographs (1037 micrographs did not contain any TOROID filaments), resulting in 8625 picked filaments.'

Fig. S3 lists 218872 particles selected for non-uniform refinement, but Supp. Table 1 lists 213808 particles finally used during processing.

This is a typo in Supplementary Table 1, the final deposited map had a corresponding particle stack of 218,872 particles, not 213,808 (this number corresponds to the particle stack of an earlier 3D refinement where more stringent particle selection parameters were used). This is changed accordingly in Supplementary Table 1.

For real-space refinement into a D1 symmetric map, have the authors possibly used NCS constraints instead of NCS restraints?

Yes, this is corrected in the revised main text.

Although the helical nature of filaments should provide a full set of views, please add a 3D-FSC and viewing direction plot to Fig.S3 for the single-particle reconstruction.

These plots are now added to Supplementary Figure 3 (panel e). As expected, the Azimuth/Elevation Orientation Distribution Plot displays a broad and even coverage of particle orientations.

In the caption to Figure S3, is (b) and (c) mixed up? (b) should be “helical reconstruction” and (c) SPA.

Yes, the figure legend of Supplementary Figure 3 is corrected accordingly.

In the local resolution map for single particle reconstruction, the drop in local resolution at the lateral ends is extreme (and may even look less so based on the box sizes used to calculate local resolution), despite an apparently large interaction interface in this direction in the TOROID filament. Is this only due to conformational variability or could the wall of the filament have holes or missing protomers? Did the authors try to use 3D-classification or variability analysis to identify subsets of the approx. 200'000 particles that are more homogenous in this region?

This drop in local resolution at the lateral ends left and right of the central TORC1 is caused by the mask we used during signal subtraction. We chose to take a smooth mask that is broad enough to completely ‘cover’ the intra-coil interface between adjacent TORC1 subunits, so that this region would be interpretable, but the part of the TORC1 subunit preceding (or succeeding) the interface with the central TORC1 is essentially cut by the used mask.

3D classification was tried, but did not result in distinctive 3D classes.

Page 9 top: “constitutively from TOROIDS” → “constitutively form TOROIDS”

This is changed in the main text.

Page 11: add closing parentheses after Pib2.

This is changed in the main text.

Reviewer #2:**Remarks to the Author:**

In this study Loewith and colleagues report on a follow-up from their earlier work, in which they had reported the formation of large assemblies of TORC1 referred to as TOROIDs in glucose-deprived yeast cells. Here the authors provide a greatly improved cryoEM structure of TOROID (at about 4Å resolution) that is combined with functional yeast genetic and imaging data to propose a novel model for the regulation of TOROID assembly and dissolution via the EGO complex. They use elegant yeast genetics to demonstrate that acute removal of EGO from the yeast vacuole induces TOROID formation and this correlates with impaired TORC1 activity evidenced by hypersensitivity to Rapamycin. Live imaging of endogenously tagged fluorescent Kog1/ Raptor and Ego3 suggests that EGO partially colocalizes with TOROID but EGO condensates are not required for TOROID assembly. Instead, it is confirmed that TOROIDs are largely controlled by Gtr/ Rag activity status; i.e. active Gtr/EGO counteracts TOROID formation, while inactive Gtrs promote TOROID assembly. The authors further present a high resolution 3D reconstruction of TOROID, in which the dimeric TORC1 building blocks appear to adopt an inactive conformation reminiscent of inactive mammalian mTORC1. Within the TOROID assembly intra-coil interactions between two adjacent TORC1 dimers can be traced to extensive interactions between Kog1/ Raptor in one and Tor2 in the other TORC1 unit. The main interface capitalizes on cation- π interactions via conserved Trp residues in Tor2 and a basic patch in Kog1. Surprisingly, the only unique elements (i.e. by comparison to mammalian Raptor) identified in Kog1 called the Twix region and the Tack helix are dispensable for TOROID assembly or binding to EGO. Instead, they may contribute to occlusion of the active site in Tor when assembled into TOROID. Finally, Raptor/ Kog1 swap mutants are combined with random mutagenesis to define an interface comprising helices 21,24,26, and 29 of yeast Kog1 that are involved in association with the EGO complex. Based on these results the authors propose that EGO modulates TOROID dynamics and thereby TORC1 activity in cells via Kog1 binding.

The data contained in this manuscript are of high quality and of interest to the community. Some of the findings are poorly supported by data as figures 3D, 7G and S7 (as quoted in the text) appear to be missing from the submission and thus render a proper assessment of this study difficult. I also found the paper hard to follow at times and noted an unusually large number of typographical errors, open brackets etc. These latter shortcomings can be corrected and improved in a suitably revised version of this paper. In addition, a number of important scientific questions remain to be resolved before publication of this work.

Response to reviewer:

We appreciate that the reviewer finds our work to be of high quality and of interest to the community. We acknowledge that the presentation of our results can be improved. Our revised manuscript is indeed much shorter, typographical errors have been corrected and the text has been edited for improved clarity.

Reference to Figure 3D on page 6: This was a typo, replaced by Figure 3C in the main text.

Reference to Figure 7G on, page 10: This was a typo, replaced by Figure 7E in the main text.

Reference to Figure S7 on page 10: Figure S7 appears to be missing in the submitted pdf. We apologize for this, and have included Figure S7 in the revised version.

Major issues:

1. A key structural element of the study is the presence of unique intra-coil interactions between Kog1 and all other subunits of the adjacent monomer within the dimeric unit including the kinase domain itself. However, from the data presented it is unclear to me to which degree the identified interface truly renders Tor kinase inactive. A more thorough analysis of these interactions and their relationship to kinase activity is required, ideally by kinetic biochemical experiments involving mutant proteins. In this context: What is the role of the Tack helix interactions displayed in Fig 5a,ii on Tor kinase activity? As it stands, I found no real evidence for the functional importance of either the Twix or the Tack helices in the control of TORC1 function. Does TORC1 become hyperactive when either Twix or Tack are deleted/ mutated?

The Tack and Twix helices are prominent features of the TOROID structure. Their discovery was unanticipated as equivalent features are not immediately obvious in mTORC1 structures reported to date. We thus focused much effort on the functional characterization of these features. The Twix mutant has been phenotypically characterized for its sensitivity to rapamycin, TOROID formation kinetics and in vivo TORC1 activation (Figure 6) and no big differences from WT cells were observed. The Tack deletion moderately compromises TOROID formation (Figure S7F). However, and as the reviewer correctly points out, neither the Twix nor Tack appear to play critical roles in TORC1 regulation. This conclusion is now more clearly articulated in the revised manuscript. Unfortunately, although we have ample material to visualize TOROIDS on grids, having enough, homogeneous material with which to perform reliable biochemical assays remains out of reach for us. In this particular case, from our in vivo results, we anyways would not expect to observe big differences from wt TORC1/TOROIDS. Respectfully, we have thus chosen/are not able to not include additional biochemical assays of TORC1 activity.

2. Both active and inactive EGOC are shown to associate with TORC1 and/or TOROID to different degrees. The authors presume that the Kog1 interface identified through mutagenesis associates with Gtr/Rags but no direct evidence is provided. Furthermore, the authors speculate in the discussion that EGOC may bind to TORC1 via multiple modes. Is the same Kog1 interface used to associate with both active and inactive EGOC? What is the explanation for the enhanced binding of inactive Gtr/EGOC to TORC1 and TOROID compared to the weak association of active EGOC with TORC1?

*We include data showing that TORC1 containing kog1^{HC} or kog1^{Δα21} lose the ability to interact with active EGOC but retain the ability to interact with inactive EGOC (FigS7E). These data indicate that different binding surfaces are used to engage these two EGOC conformers. In the absence of structures of TORC1/TOROID – EGOC^{on/off} complexes (which we hope the reviewer agrees is beyond the scope of this current submission) we are reluctant to speculate further on the observed differences in relative binding strengths/modes. We note in the discussion that loss of EGOC activity in *S. pombe* results in hyperactivation of TORC1, consistent with the notion that the EGOC also serves to inhibit TORC1 in some conditions. Addressing the positive and negative regulation of the EGOC on TORC1 will be a focus of our future efforts.*

3. The believed to be critical pi-cation interaction between Tor and Kog1 is only addressed by mutation of the two Trp in Tor while mutation of a single Arg in Kog1 appears to be without effect. Further biochemical evidence for the charge complementation model, e.g. by analysis of mutants of the entire basic patch would greatly strengthen the model and increase the impact of the study.

We compared TOROID formation in GFP-TOR1 WT and GFP-TOR1 kog1R884D cells. In exponential phase, we do not observe any difference however, PDS we observe a 30 % decrease (WT: 72.7 +/- 0.03, mutant: 52 +/- 0.02), mentioned in the revised manuscript as data not shown.

Despite multiple attempts, we unfortunately have been unable to generate a kog1 mutant wherein the entire basic patch has been targeted (R884A/R885A/K888A). We note that mutation of this surface in Kog1 will likely also effect EGOC binding which would anyways complicate interpretation of results.

4. The distribution of figure panels between the main figures and the supplement could be improved. For example, Fig. S4C appears to be quite central and should be displayed as part of a main figure.

A new version of Figure 4 includes this request.

Reviewer #3:**Remarks to the Author:**

The manuscript is an important contribution to understanding regulation of TOR. Prouteau et al. have described in unprecedented detail the structure of a large assembly of budding yeast TORC1 units that has been described as the TOROID. A previous report of this structure at low resolution was sufficient to gain a general impression of the arrangements of the units within the filaments that make up the TOROID. However, in the current manuscript they have improved the cryo-EM structure by doing a focused refinement with subtraction for each TORC1 subunit with its contacting units within the TOROID. Aligning all of these units gives a 3.8 Å resolution image of the intra-coil interactions. They conclude that the contacts within the TOROID block the active site of the TORC1 through interactions with the FRB, similarly to how FKBP/Rapamycin interacting with the FRB inactivates the enzyme. Their cellular observations are consistent with this being an inactive state of the TORC1, since light microscopy suggests formation of TOROID-containing puncta when glucose is restricted and TORC1 activity is downregulated.

In addition to the structural observations, the authors have also shown that the active EG1/2/3/Gtr1/2 (EGOC) complex interacts with TOROIDS, and this is consistent with the general picture that Loewith and his colleagues developed prior to this study: active Gtrs bind TORC1 and disassemble TOROIDS, because the surface on Kog1 that interacts with active Gtrs is a region that encompasses interactions between Kog1 and the TOR from an adjacent subunit in the TOROID. The current higher-resolution work fills in the molecular details of these interactions and verifies the structure by mutagenesis. However, it also shows that the picture is incomplete, because they convincingly show that the inactive EGOC also robustly binds TORC1. They propose that TORC1 is capable of making two different types of interactions with EGOC, depending on the nucleotide state of the Gtrs. Although mTORC1 assemblies analogous to TOROIDS have not been reported, it may be that similar assemblies occur in circumstances that have not yet been characterized. Such assemblies could give an additional layer of control to mTORC1. The manuscript will provide a useful framework for designing experiments to examine the possibility that mTORC1 could also form TOROID-like assemblies.

The yeast work is elegant and convincingly executed to illustrate the regulation of TOROIDS by EGOC and the glucose state. The structural work is innovative and informative.

Response to reviewer:

We thank the reviewer for his/her constructive and enthusiastic feedback. Specific responses to the reviewer's comments follow below.

There are some minor points that should be addressed in revision. These are summarized in the following.

lines 133-134:

The authors start the results with “We previously proposed that TOROID regulation downstream of glucose-derived signals was mediated via the EGO (27)” The word “regulation” is sloppy and should be replaced. Is it up-regulation or down-regulation? Does EGO enhance TOROIDS or repress TOROIDS? Reference 27 does make it clear what they proposed EGO is doing (though it too is plagued with similar ambiguous writing). The sentence should be re-written to make it clear: We proposed that EGO interaction prevents formation of TOROIDS.

We have taken the comment in account and rephrased the sentence.

Lines 90-93:

The authors state “TOROID formation was found to be dependent upon the two yeast RAG orthologues GTR1 (RAGA/B) and GTR2 (RAGC/D): delta gtr1 /delta gtr2 cells present constitutive TOROIDS even in the presence of glucose and are impaired in their ability to downregulate Sch9 phosphorylation upon glucose starvation.” The statement might be clear to geneticists or yeast biologists, but, given the journal, it should be rephrased to be a bit more accessible. The statement “TOROID formation was found to be dependent upon the two yeast RAG orthologues GTR1 (RAGA/B) and GTR2 (RAGC/D)...” sounds like the Gtrs promote assembly of TOROIDS, but active Gtrs, in fact, seem to prevent the assembly. The statement is technically correct. If Gtr1/2 prevents assembly, assembly is dependent on whether there is Gtr1/2 or not. However, it would be a little easier to follow if the authors state something like: “TOROID formation was found to be inhibited by the two yeast RAG orthologues GTR1 (RAGA/B) and GTR2 (RAGC/D), because delta gtr1 /delta gtr2 cells form constitutive TOROIDS, even in the presence of glucose and do not downregulate Sch9 phosphorylation upon glucose starvation.”

We have tried our best to rewrite the sentence in a more comprehensible manner. Given that, depending on nucleotide loading, the EGO appears to both positively and negatively regulate TORC1 activity, we need to be careful when describing the “regulation” of TORC1 by the EGO, as the reviewer points out below.

Lines 107-110:

The sentence “Indeed, expressing mutants that lock the Gtrs in an inactive, Gtr1GDP/Gtr2GTP conformation, TORC1 is concentrated in a TOROID punctum on the

vacuolar membrane whereas in cells expressing active, Gtr1GTP/Gtr2GDP, TORC1 is diffusely localized over the surface of the vacuole 27.”
 Would read better as: “Indeed, by expressing mutants that lock the Gtrs in an inactive, Gtr1GDP/Gtr2GTP conformation, TORC1 is concentrated in a TOROID punctum on the vacuolar membrane whereas by expressing active, Gtr1GTP/Gtr2GDP, TORC1 is diffusely localized over the surface of the vacuole 27.”(insert the word by)

We have taken the comment in account and rephrased the sentence.

Line 121:

The authors state “We found that a central hub of interactions locks the Kog1 Armadillo (ARM) domain on top of the kinase domain of Tor2’and sterically occludes the kinase active site.”

This term Tor2’ has not yet been introduced in the text.

I would suggest “We found that a central hub of interactions locks the Kog1 Armadillo (ARM) domain on top of the kinase domain of the next Tor2 molecule in the TOROID (Tor2’) to sterically occlude the kinase active site.”

We have taken the comment in account by rephrasing the sentence and mentioning clearly that subunits of neighboring TORC1 are mentioned by a “prime”.

Lines 124-125 :

The authors say “this same hub is engaged by the EGOC to mediate glucose-dependent regulation of TORC1 signalling.”

I understand that regulation is a neutral term. It could mean activate or inactivate. The text would be much more dynamic and readable, if the authors take a position and say what they mean. I would suggest:

“this same hub is engaged by EGOC to activate TORC1 signalling in response to glucose.”

We have taken the comment in account and rephrased the sentence.

Lines 125-126:

“we propose that the EGOC is necessary and sufficient for TOROID regulation through competition for a common binding hub in Kog1”

Avoiding the neutral, boring and less informative word “regulation”, it would be better to say something like:

we propose that the EGOC is necessary and sufficient to prevent TOROID assembly, by competing for a common binding hub in Kog1.

We have taken the comment in account and rephrased the sentence.

lines 127-129:

“trans binding of this hub to Tor1’/2’ sequesters TORC1 into an inactive TOROID, while binding to active EGOC liberates TORC1 dimers able to signal to downstream targets.” Again, the authors are using a term Tor1’/Tor2’ that has not yet been introduced. What does “trans” binding mean? This has not been defined in the text at this point.

We have taken the comment in account by removing the term “trans”, rephrased the sentence and precise how we mention subunits of neighboring TORC1.

In Fig 3, the domain that the authors refer to as the FAT appears to be what has been referred to as the M-HEAT or bridge in previous papers. The region referred to as the kinase domain appears to be the FAT and kinase domain (also known as the FATKIN). The authors should use the FAT and kinase domain consistent with previous publications.

This labeling mistake has been corrected.

In Fig. 4 and in text lines 283 and 284. mTORC1 without RHEB is referred to as “inactive”. This is misleading. The enzyme activity of this complex without has been characterized by several groups. It is more appropriate to refer to this as basal state or apo-state.

The authors should state clearly how they measure the angles illustrated in Figs 4A,B. There appears to be nothing in the methods.

We have taken the comment into account and refer now to apo-state instead of active. A description of how the angles were measured has been included in the figure legend.

In Fig 3 and lines 295,296, the authors refer to Tor2’. Is this Tor2’ in an adjacent unit in the Toroid, or is it the other Tor2 in the Torc2 dimer?

It would be unambiguous if the authors change “...interacts with the FRB’ domain of Tor2’ in an adjacent TORC1’ subunit. “ to It would be unambiguous if the authors change “...interacts with the FRB’ domain of Tor2’ in an adjacent TORC1’ in the TOROID. “ The confusion arises because TORC1 is not a subunit. It is a complex.

We have taken the comment in account and rephrased the sentence.

Lines 311-313 :

“The Twix and the Tack shield the kinase active site and thereby block substrate access, providing, together with the large allosteric change described above, a structural

explanation for why TORC1 in TOROIDs is inactive (Figure 4D, Figure S4C).]” First. There should be a clear picture of where they would expect a substrate peptide interacting with the FRB/Kog1 interface in the TOROID. There is a region circled called “active site”, but no explanation as to what this is or how it is deduced. In Fig S4C, there is a region colored green in the active site, but the legend does not explain what this is. Is this the ATP-binding site or the peptide-binding site on the FRB? Second. There has been no allosteric change described above. The only allosteric change described above is the conformational change between apo and RHEB-bound mTORC1 conformations. There is a statement that the TOROID is more open than the mTORC1 dimer, but this is not an allosteric change. The proper comparison to infer allostery would be to the TORC1 dimer.

Unfortunately, the exact mode of binding of a peptide with the yeast TORC1 is not known, nor has a yeast TORC1 dimer structure has not been reported to perform the requested comparison to determine allostery changes. We have indicated in the modified Figure 4D where ATP binds based on mTORC1 structures. In addition, we have toned down our conclusion to this section:

The Twix and the Tack shield the kinase active site and thereby block substrate access, potentially contributing (see below), together with the presumed TORC1 subunit allosteric change described above, a structural explanation for why TORC1 is inactive in TOROIDs (Figure 4D).

Lines 463-465 :

“Replacement of the proximal alpha-helix 34 (Kog1 1004-1022) had no effect on TOROID regulation, while replacement of the entire Tack helix (Kog1 1121-1131) leads to a mild, but significant, impairment of TOROID formation (Figure S7E, F).” There appears to be a typo here. The Tack helix is 1069-1086. Not 1121-1131 (this is the claw).

We thank the reviewer to pick up on this mistake. We corrected it.

Lines 497-498 :

“a dynamic equilibrium between active, free TORC2 dimers 497 and inactive TOROID polymers in a cell”

This is probably a typo. TORC2 should be TORC1. This is the only place in the manuscript that TORC2 has been mentioned.

Decision Letter, first revision:

Message: Our ref: NSMB-A45852A

27th Sep 2022

Dear Dr. Loewith,

Thank you for submitting your revised manuscript "EGOC inhibits TOROID polymerization by structurally activating TORC1" (NSMB-A45852A). It has now been seen by two of the original referees and their comments are below. The reviewers find that the paper has improved in revision, and therefore we'll be happy in principle to publish it in Nature Structural & Molecular Biology, pending minor revisions to satisfy the referees' final requests and to comply with our editorial and formatting guidelines.

Kind regards,
Florian

Dr Florian Ullrich
Associate Editor, Nature
Consulting Editor, Nature Structural & Molecular Biology
ORCID 0000-0002-1153-2040

Reviewer #1 (Remarks to the Author):

The authors have adequately addressed the points raised by this reviewer.

Have the authors changed or updated the map/model used ? Without a clear mentioning of changes in the response to reviewers, the resolution was changed in the text and supplementary table 1 from 3.85A to 3.87A (line 207) (and from 3.8A to 3.9A in the abstract), while the FSC plot in suppl. figure S3d still shows 3.85A and this number is also reported in figure S3a (same inconsistency for updated B factor of -131.8). It looks to me like Table S1 and text where updated for a new final reconstruction/refinement, but the suppl. fig. wasn't. Please, check carefully for consistency.

Comments on Responses to Reviewer 2:

Reviewer 2 requested a clear discussion of the role of key interaction regions in TOROIDs, coined Twix and Tack. In their response to the reviewer, the authors write: "However, and as the reviewer correctly points out, neither the Twix nor Tack appear to play critical roles in TORC1 regulation. This conclusion is now more clearly articulated in the revised manuscript."

As both elements function in one interaction hub that is declared to block substrate access in TOROIDs, a corresponding conclusive summary statement should be added to the discussion, together with a concise discussion of what this means for the role of TOROIDs. Currently the information is partially present, but spread out over different paragraphs, and the discussion section doesn't mention Tack at all.

cf.

244: hub ... near Tack α -helix and the base of the Twix region ... interactions within this hub shield the kinase active site and thereby block substrate access

303: Twix helices are not necessary for TOROID formation or TORC1 regulation

362: Tack helix provides structural elements needed to stabilize the TOROID

... In exponential phase, we do not observe any difference however, PDS we observe a 30 % decrease (WT: 72.7 +/- 0.03, mutant: 52 +/- 0.02), mentioned in the revised manuscript as data not shown.

=> Please, show data in suppl. figure.

Comments on Responses to Reviewer 3:

Referring to mTORC1 without Rheb as apo-state: I rather suggest to use the other suggestion provided by the reviewer, basal (or non-activated) state. Rheb is an activating and transiently binding protein factor while the term pair apo/holo typically refers to a prosthetic group or more permanently interacting cofactor contributing directly to catalysis.

Reviewer #2 (Remarks to the Author):

My previous comments have been addressed.

Decision Letter, author guidance:

Message: Our ref: NSMB-A45852A

7th Oct 2022

Dear Dr. Loewith,

Thank you for your patience as we've prepared the guidelines for final submission of your

Nature Structural & Molecular Biology manuscript, "EGOC inhibits TOROID polymerization by structurally activating TORC1" (NSMB-A45852A). Please carefully follow the step-by-step instructions provided in the attached file, and add a response in each row of the table to indicate the changes that you have made. Please also check and comment on any additional marked-up edits we have proposed within the text. Ensuring that each point is addressed will help to ensure that your revised manuscript can be swiftly handed over to our production team.

We would like to start working on your revised paper, with all of the requested files and forms, as soon as possible. If you can resubmit within the next week it is possible that your submission could be published before the end of 2022. Please get in contact with us if you anticipate any delays in resubmission.

In recognition of the time and expertise our reviewers provide to Nature Structural & Molecular Biology's editorial process, we would like to formally acknowledge their contribution to the external peer review of your manuscript entitled "EGOC inhibits TOROID polymerization by structurally activating TORC1". For those reviewers who give their assent, we will be publishing their names alongside the published article.

Nature Structural & Molecular Biology offers a Transparent Peer Review option for new original research manuscripts submitted after December 1st, 2019. As part of this initiative, we encourage our authors to support increased transparency into the peer review process by agreeing to have the reviewer comments, author rebuttal letters, and editorial decision letters published as a Supplementary item. When you submit your final files please clearly state in your cover letter whether or not you would like to participate in this initiative. Please note that failure to state your preference will result in delays in accepting your manuscript for publication.

Cover suggestions

As you prepare your final files we encourage you to consider whether you have any images or illustrations that may be appropriate for use on the cover of Nature Structural & Molecular Biology.

Nature Structural & Molecular Biology has now transitioned to a unified Rights Collection system which will allow our Author Services team to quickly and easily collect the rights and permissions required to publish your work. Approximately 10 days after your paper is formally accepted, you will receive an email in providing you with a link to complete the grant of rights. If your paper is eligible for Open Access, our Author Services team will also be in touch regarding any additional information that may be required to arrange payment for your article.

Please note that *Nature Structural & Molecular Biology* is a Transformative Journal (TJ). Authors may publish their research with us through the traditional subscription access route or make their paper immediately open access through payment of an article-processing charge (APC). Authors will not be required to make a final decision about access to their article until it has been accepted. [Find out more about Transformative Journals](https://www.springernature.com/gp/open-research/transformative-journals)

Authors may need to take specific actions to achieve [compliance with funder and institutional open access mandates](https://www.springernature.com/gp/open-research/funding/policy-compliance-faqs). If your research is supported by a funder that requires immediate open access (e.g. according to [Plan S principles](https://www.springernature.com/gp/open-research/plan-s-compliance)) then you should select the gold OA route, and we will direct you to the compliant route where possible. For authors selecting the subscription publication route, the journal's standard licensing terms will need to be accepted, including [self-archiving policies](https://www.nature.com/nature-portfolio/editorial-policies/self-archiving-and-license-to-publish). Those licensing terms will supersede any other terms that the author or any third party may assert apply to any version of the manuscript.

Please use the following link for uploading these materials:
[Redacted]

Best regards,

Sophia Frank
Editorial Assistant
Nature Structural & Molecular Biology
nsmb@us.nature.com

On behalf of

Florian Ullrich, Ph.D.
Associate Editor
Nature Structural & Molecular Biology
ORCID 0000-0002-1153-2040

Reviewer #1:

Remarks to the Author:

The authors have adequately addressed the points raised by this reviewer.

Have the authors changed or updated the map/model used ? Without a clear mentioning of changes in the response to reviewers, the resolution was changed in the text and supplementary table 1 from 3.85A to 3.87A (line 207) (and from 3.8A to 3.9A in the abstract), while the FSC plot in suppl. figure S3d still shows 3.85A and this number is also reported in figure S3a (same inconsistency for updated B factor of -131.8). It looks to me like Table S1 and text where updated for a new final reconstruction/refinement, but the suppl. fig. wasn't. Please, check carefully for consistency.

Comments on Responses to Reviewer 2:

Reviewer 2 requested a clear discussion of the role of key interaction regions in TOROIDS, coined Twix and Tack. In their response to the reviewer, the authors write: "However, and as the reviewer correctly points out, neither the Twix nor Tack appear to play critical roles in TORC1 regulation. This conclusion is now more clearly articulated in the revised manuscript."

As both elements function in one interaction hub that is declared to block substrate access in TOROIDS, a corresponding conclusive summary statement should be added to the discussion, together with a concise discussion of what this means for the role of TOROIDS. Currently the information is partially present, but spread out over different paragraphs, and the discussion section doesn't mention Tack at all.

cf.

244: hub ... near Tack α -helix and the base of the Twix region ... interactions within this hub shield the kinase active site and thereby block substrate access

303: Twix helices are not necessary for TOROID formation or TORC1 regulation

362: Tack helix provides structural elements needed to stabilize the TOROID

... In exponential phase, we do not observe any difference however, PDS we observe a 30 % decrease (WT: 72.7 +/- 0.03, mutant: 52 +/- 0.02), mentioned in the revised manuscript as data not shown.
=> Please, show data in suppl. figure.

Comments on Responses to Reviewer 3:

Referring to mTORC1 without Rheb as apo-state: I rather suggest to use the other suggestion provided by the reviewer, basal (or non-activated) state. Rheb is an activating and transiently binding protein factor while the term pair apo/holo typically refers to a prosthetic group or more permanently interacting cofactor contributing directly to catalysis.

Reviewer #2:
Remarks to the Author:
My previous comments have been addressed.

Final Decision Letter:

Message 21st Nov 2022

:

Dear Robbie,

We are now happy to accept your revised paper "EGOC inhibits TOROID polymerization by structurally activating TORC1" for publication as a Article in Nature Structural & Molecular Biology.

As soon as your article is published, you can generate your shareable link by entering the DOI of your article here: http://authors.springernature.com/share. Corresponding authors will also receive an automated email with the shareable link

Your paper will be published online soon after we receive proof corrections and will appear in print in the next available issue. You can find out your date of online publication by contacting the production team shortly after sending your proof corrections. Content is published online weekly on Mondays and Thursdays, and the embargo is set at 16:00 London time (GMT)/11:00 am US Eastern time (EST) on the day of publication. Now is the time to inform your Public Relations or Press Office about your paper, as they might be interested in promoting its publication. This will allow them time to prepare an accurate and satisfactory press release. Include your manuscript tracking number (NSMB-A45852B) and our journal name, which they will need when they contact our press office.

About one week before your paper is published online, we shall be distributing a press release to news organizations worldwide, which may very well include details of your work. We are happy for your institution or funding agency to prepare its own press release, but it must mention the embargo date and Nature Structural & Molecular Biology. If you or your Press Office have any enquiries in the meantime, please contact press@nature.com.

Please note that *Nature Structural & Molecular Biology* is a Transformative Journal (TJ). Authors may publish their research with us through the traditional subscription access route or make their paper immediately open access through payment of an article-processing charge (APC). Authors will not be required to make a final decision about access to their article until it has been accepted. <https://www.springernature.com/gp/open-research/transformative-journals> Find out more about Transformative Journals

Kind regards,
Florian

Dr Florian Ullrich
Associate Editor, Nature
Consulting Editor, Nature Structural & Molecular Biology
ORCID 0000-0002-1153-2040